# Scaffold-enabled high-resolution cryo-EM structure determination of RNA

Daniel B. Haack [1], Boris Rudolfs[1], Shouhong Jin[2], Alexandra Khitun[2], Kevin M. Weeks [2] & Navtej Toor [1] ✉

Cryo-EM structure determination of protein-free RNAs has remained difficult with most attempts yielding low to moderate resolution and lacking nucleotide-level detail. These difficulties are compounded for small RNAs as cryo-EM is inherently more difficult for lower molecular weight macro-molecules. Here we present a strategy for fusing small RNAs to a group II intron that yields high resolution structures of the appended RNA. We demonstrate this technology by determining the structures of the 86-nucleotide (nt) thiamine pyrophosphate (TPP) riboswitch aptamer domain and the recently described 210-nt *raiA* bacterial non-coding RNA involved in sporulation and biofilm formation. In the case of the TPP riboswitch aptamer domain, the scaffolding approach allowed visualization of the riboswitch ligand binding pocket at 2.5 Å resolution. We also determined the structure of the ligand-free apo state and observe that the aptamer domain of the riboswitch adopts an open Y-shaped conformation in the absence of ligand. Using this scaffold approach, we determined the structure of *raiA* at 2.5 Å in the core. Our versatile scaffolding strategy enables efficient RNA structure determination for a broad range of small to moderate-sized RNAs, which were previously intractable for high-resolution cryo-EM studies.

There is great interest in the development of small molecules targeting RNA structures for the treatment of disease. The design of small molecule drugs that target proteins is notably accelerated by studying structure activity relationships (SAR) to optimize binding to protein targets, as informed by routine visualization of protein-ligand complexes. However, structure-informed SAR is currently difficult and rare for the development of small molecules targeting RNA structures. Most RNAs do not crystallize readily for x-ray crystallography and also present difficulties for cryo-electron microscopy (cryo-EM) structure determination. In our experience, vitrification on cryo-EM grids often results in the denaturation and aggregation of RNA[1] and often only a small number of native particles are visualized, which is insufficient to obtain a high-resolution reconstruction. In addition, the few particles that are visualized often exhibit a preferred orientation such that the views of the particle are not sufficiently populated for high resolution 3D reconstruction. Therefore, there is a compelling need for

technologies to overcome these obstacles and enable routine high-resolution cryo-EM analysis of protein-free RNAs. Obtaining high-resolution structures of RNA and RNA-ligand complexes is crucial for establishing molecular mechanisms of action in multiple biological contexts.

Cryo-EM has revolutionized structural biology with thousands of protein structures solved and shared in the Protein Data Bank (PDB). In contrast, there has only been a single protein-free RNA (*Tetrahymena* group I intron) determined at 3 Å resolution or better using cryo-EM[2]. The remainder of current protein-free RNAs have been solved to resolutions ranging from 5 to 10 Å[3], suggesting that RNA is not as amenable to cryo-EM structure determination as are proteins. At 5 Å resolution, there are two major problems: de novo modeling is not possible, especially in complex helical junctions, and sequence register cannot be determined due to the lack of nucleobase separation. It is possible to place computationally derived RNA structures into 5 Å

[1]Department of Chemistry and Biochemistry, University of California, San Diego, CA, USA. [2]Department of Chemistry, University of North Carolina, Chapel Hill, NC, USA. ✉e-mail: ntoor@ucsd.edu

density, however, these methods currently do not accurately model complex regions. For example, the glycine riboswitch[3] and a tRNA-like viral structure[4] were reported at 5.7 and 4.3 Å, respectively. In each case, examination of the model fit to density shows an inability to accurately determine sequence register, with nucleotides modeled outside of density (Fig. S1).

One approach for solving RNA structures via cryo-EM is to use a scaffold to which an RNA motif of interest is appended. The goal is that the favorable biophysical properties that result in high resolution for the scaffold would propagate to the target RNA. The larger mass of the overall RNA also facilitates particle picking and alignment for 3D reconstruction. This approach has been attempted with the *Tetrahymena* group I intron, in multiple adaptations[5,6]. This group I intron, in isolation, has been solved to ~3 Å resolution[2]. However, to date, this approach has resulted in only ~5 Å resolution for the target RNA[5,6], which does not allow for the discrimination of individual nucleotides and makes precise modeling impossible (Fig. S1). For example, when the group I intron was used as a scaffold to determine the structures of the fluoride riboswitch and the Zika virus xrRNA, modeling these RNA structures required the use of prior high-resolution crystal structures. At this resolution, it was not possible to perform de novo modeling (zoomed-in insets in Fig. S1), as noted previously[6]. The scaffold approach has also been attempted using RNA origami[7] and the ribosome[8], with similar results. In summary, current scaffolding approaches have not yet demonstrated the ability to achieve high resolution structure determination of attached target RNAs that allows discrimination of individual nucleobases. Further development of technologies that would allow routine application of cryo-EM to protein-free RNAs would have a large impact on RNA structural biology and for development of small molecules targeting RNA.

X-ray crystallography can be applied for the structure determination of RNA, but has limitations. It is challenging to form crystals with protein-free RNAs due to the relatively homogeneous negatively charged surface, making it difficult to form unique crystal contacts. RNA conformational dynamics are also not fully assessed in a crystal structure because the constraining crystal lattice packs macromolecules together at concentrations approaching hundreds of mg/ml, suppressing helical dynamics and favoring compact states. For example, in our structural studies of group II introns, we attempted to characterize branch-site helix dynamics using crystallography and observed only small scale movements in the domain VI helix of a group IIB intron[9]. We subsequently determined the cryo-EM structure of a homologous group IIB intron, with a shared secondary structure, and found that the branch-site helix engages in a 90° swinging action between the two steps of splicing[10]. These outcomes emphasize that cryo-EM is better suited to providing insight into RNA dynamics.

We hypothesize that similar conformational dynamics could be observed for riboswitches if cryo-EM were feasible for these small RNAs. Riboswitches are RNA structures found in mRNAs that bind small molecule ligands to affect gene expression by altering either transcription or translation of a downstream gene. This regulation occurs through the sequestering or releasing of a regulatory RNA element[11]. Most riboswitches have two domains, an aptamer domain which binds the regulating ligand and forms an RNA motif with a specific higher-order tertiary structure, and an expression domain whose formation, or not, regulates either transcription or translation of the downstream gene. The riboswitch mechanism implies there is an equilibrium between different conformations, with the presence or absence of ligand shifting this equilibrium to govern the resulting effects upon gene expression[12–14]. However, crystal structures of most riboswitches typically show minor differences between the bound and apo states[15–20]. As in the case of the group II intron, this similarly is likely due to the constraining environment of the crystal lattice.

In addition to riboswitches, there are large classes of other non-coding RNAs (ncRNAs) that have long been known to have important biological roles in both prokaryotes and eukaryotes[21]. These newly discovered bacterial ncRNAs are predicted to have ordered RNA structures based on phylogenetically conserved secondary structures[21]. One recent example is the *raiA* non-coding RNA motif that is found in over 2500 bacterial species[22]. RaiA from *Clostridium acetobutylicum* is 210 nucleotides in length and more than 25% of its nucleotides exhibit 97% conservation, strongly indicating that the RNA plays an important biological role[22]. RaiA also contains highly-conserved nucleotides in single stranded regions whose conservation cannot be explained by the predicted secondary structure[22]. Knockouts of the gene encoding *raiA* results in defects in both bacterial sporulation and biofilm aggregation[22]. The *raiA* ncRNA is also the fourth most abundant RNA when bacterial cells transition from exponential growth to the stationary phase, and the resulting spores in *raiA* mutants exhibit 10% viability compared to wild-type (WT)[22]. The cumulative evidence suggests that *raiA* has a conserved structure that is essential for its biological function in vivo. In contrast to riboswitches and catalytic introns, ncRNAs like *raiA* remain poorly characterized in terms of their 3D structure and function. Previous attempts at structural characterization of these ncRNAs were largely limited to chemical probing to confirm their secondary structures[23]. In the absence of known functional roles for ncRNAs, it would be useful to determine the high-resolution structure from which the first hints of possible function might be gleaned.

Here, we report our development of a group II intron scaffold that allows for cryo-EM structure determination of attached small RNAs at high resolution. We tested this approach on RNA targets of both known and unknown structure. For our known structure candidate, we solved cryo-EM structures of the 86-nucleotide (nt) thiamine pyrophosphate (TPP) riboswitch aptamer domain, in both ligand-bound and ligand-free (apo) states. We visualized the ligand-binding pocket of the TPP riboswitch at 2.5 Å resolution, which enabled precise modeling of the thiamine pyrophosphate ligand. Comparisons of the ligand-bound and apo structures reveal conformational dynamics that inform a mechanism for riboswitch function. For the target of unknown structure, we determined the cryo-EM structure of the bacterial non-coding *raiA* at 3.0 Å, with the core at 2.5 Å, which allowed for the de novo modeling of the entire RNA. With these advances, it is now possible to determine the structure of an ncRNA in the beginning stages of discovery to inform the design of subsequent biochemical and biological experiments, and to visualize bound ligands. Going forward, this technology allows structure determination at sufficiently high resolution to enable structure-based SAR for small molecules that bind selectively to RNA.

## Results

### Identification of a high-resolution RNA scaffold

We sought to identify an improved scaffold to hold small target RNAs for high-resolution cryo-EM structure determination. An ideal RNA scaffold should have the following properties: (1) exhibit a minimal grid orientation preference, (2) have high solubility, (3) have a structure resistant to denaturation, (4) have a molecular mass >100 kDa, and (5) contain a bridging region that can accommodate target RNAs. An RNA with these properties should ideally result in cryo-EM density of 3 Å or better resolution.

We screened multiple group II introns for their suitability as scaffolds for cryo-EM. Group II introns are self-splicing catalytic RNAs that have six distinct domains and typically range in size from 400 to 850 nucleotides. We first tested the attachment of a group IIC intron from *Oceanobacillus iheyensis* (*O.i.*)[24] (120 kDa) to a larger (scaffolding) group IIB intron ribonucleoprotein (RNP) complex from *Thermosynecoccus elongatus* (*T.el.*)[10] (300 kDa) to form the *T.el.-O.i.* fusion construct. When local refinement was performed on the scaffold portion of the density, a 3.5 Å map of the RNP was recovered (Fig. 1A). However, when local refinement was shifted to density corresponding

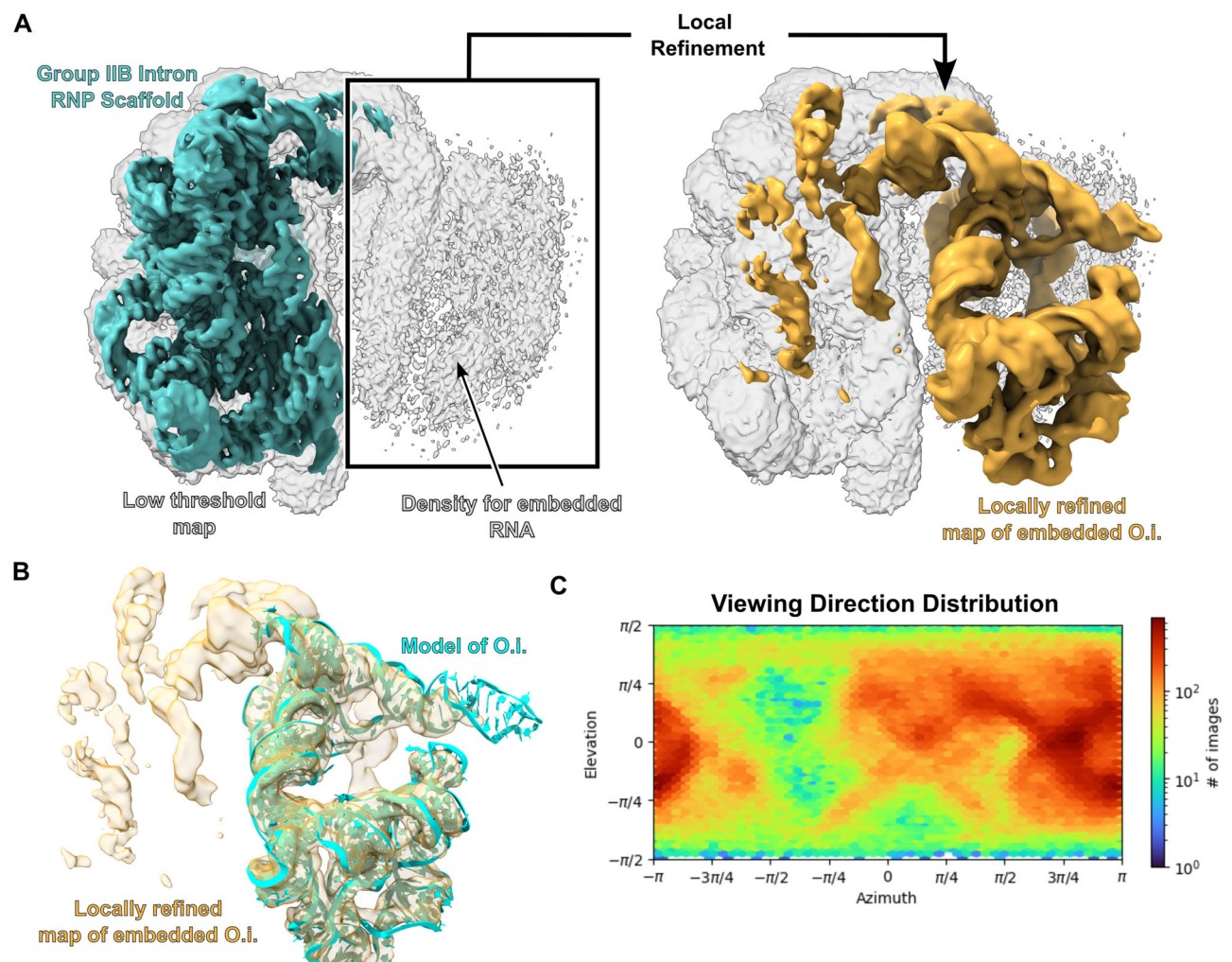

**Fig. 1 | Initial scaffold development using a group IIB intron RNP. A** The maps of the group IIB RNP were generated using a tilted data collection strategy with the stage set to 30°. The map on the left was globally refined and shown at both a high (teal) and low threshold (white). The group II intron RNP displays clear RNA features at high threshold and density for the embedded RNA is observed at low threshold. Performing local refinement on the density corresponding to the embedded RNA yielded the map on the left (gold) which has been overlaid on the low threshold globally refined map (white) for clarity. **B** A model for the embedded RNA was fit into the locally refined map. The fit shows that the overall fold of the embedded RNA was maintained. **C** The viewing direction distribution for the data set is shown. Even with a 30° stage tilt, there is still a preferred orientation for this specimen.

to the embedded IIC intron, the resolution of the target RNA was limited to ~5 Å. This resolution limit is consistent with prior experiences for scaffold-based approaches[5–8]. We were encouraged to observe that the overall fold of the embedded RNA was maintained even though the method resulted in only a moderate resolution map (Fig. 1B). This low resolution likely reflects that the RNP scaffold displays severe orientation bias, which is still present after employing a tilted data collection strategy at 30° (Fig. 1C). We then tested the ability of the group IIC intron in isolation to function as a scaffold. The *O.i.* intron (Fig. 2A) folds into a stable structure that crystallizes in a wide variety of precipitants[24]. However, this particular intron has the unwanted tendency to undergo intramolecular hydrolytic cleavage at multiple sites[25], which may also cleave an attached RNA target. We therefore used an inactive mutant in which an active site residue is mutated to abolish catalytic activity[26]. In our first attempt, we obtained a reconstruction at 2.4 Å resolution for this group II intron (Fig. 2B, C, and Table S1). In addition, this intron exhibited a very favorable lack of orientation preference (Fig. 2D) and has a stable solvent-exposed stem that could be used to embed an RNA of interest (domain III). The orientation distribution map is evenly populated, which is highly unusual for a protein-free RNA in our experience. Based on these

favorable properties, we selected this group IIC intron as a scaffold candidate for the attachment of target RNAs for high resolution cryo-EM.

### Construct design to attach target to the scaffold

It is necessary to attach the target RNA to the scaffold via a rigid helix to reduce flexibility and facilitate high-resolution reconstruction. The attachment site on the scaffold should have the following properties: (1) be solvent exposed so that the target RNA does not fold back onto the scaffold to disrupt the structure; (2) allow sequence changes and insertions that do not affect folding of the scaffold; and (3) exhibit limited flexibility. The domain III stem of the group II intron satisfies these requirements (Fig. 2A, B). In addition to the scaffold, there are also requirements for the target RNA to be amenable to attachment to the group II intron. The target RNA must contain a stem that can be modified/mutated without affecting its biochemical activity. This target stem is used to create a continuous helix with the attachment site on the scaffold. In some cases, a circular permutation of the target can enable use of a stem located deep within the RNA sequence. Ultimately, the target RNA is fused to the scaffold to form a single RNA that is then synthesized using in vitro transcription.

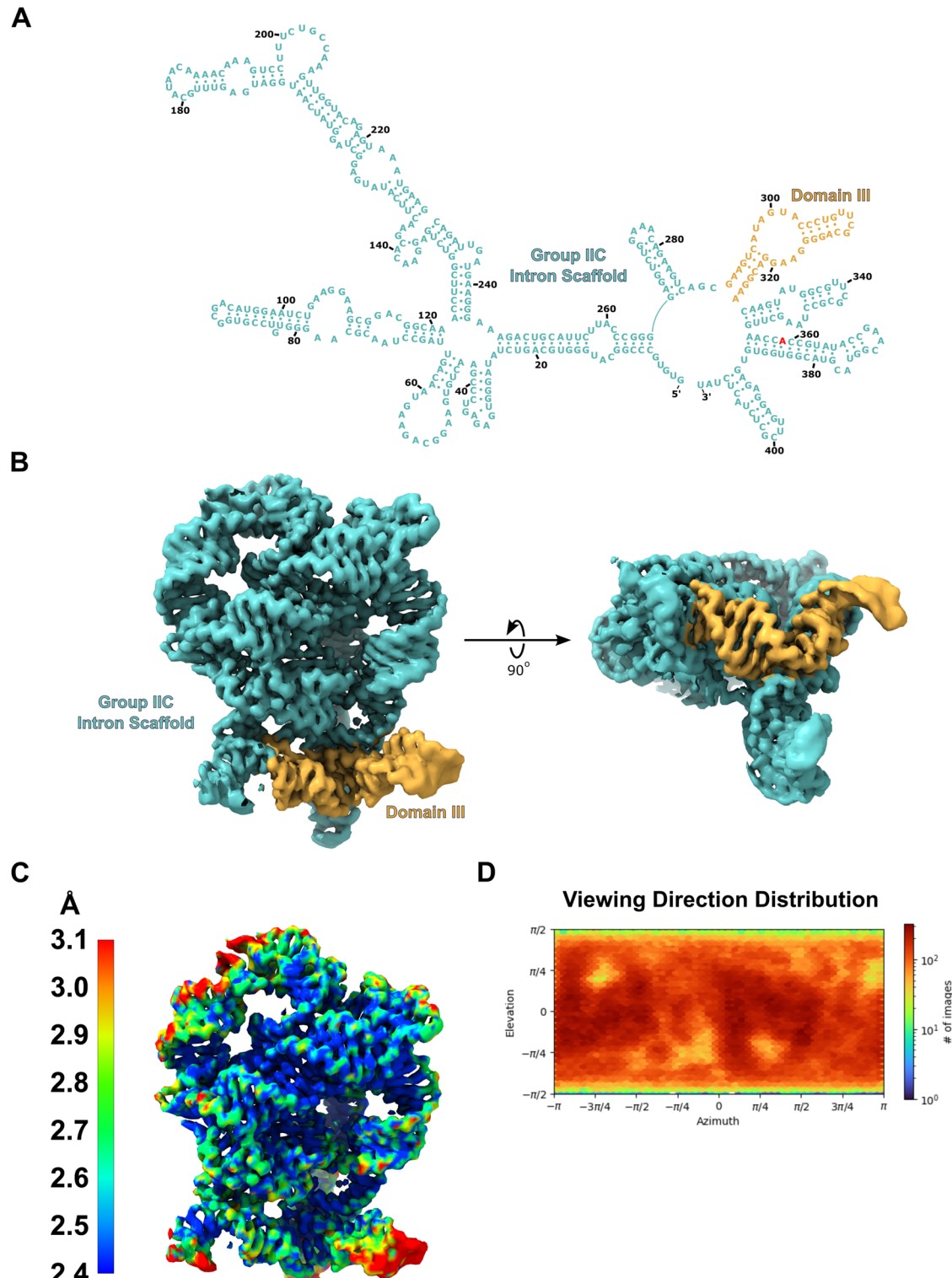

**Fig. 2 | Identification of a high-resolution group IIC intron scaffold. A** The secondary structure of the *O.i.* group IIC intron scaffold (teal) is shown highlighting domain III (gold). The G to A mutation in the active site is highlighted red. **B** Cryo-EM density is shown for the scaffold. Domain III (gold) forms a stable stem loop that extends away from the rest of the folded RNA. **C** A local resolution map shows that the core of the scaffold has a resolution of 2.4 Å. **D** The viewing direction distribution plot generated from cryoSPARC is shown. The plot is evenly populated indicating that the group IIC intron scaffold exhibits no preferred orientation.

The fusion construct is directly purified from the in vitro transcription reaction using diafiltration. Before diafiltration, the transcription reaction is first treated with DNase to degrade the plasmid DNA template and then with proteinase K to cleave T7 RNA polymerase and DNase. This solution is then subjected to buffer exchange (molecular weight cut-off of 100 kDa). For some RNAs, this procedure results in the formation of hydrogels due to the high concentration gradient that builds up on the membrane. In these cases, size exclusion

chromatography with a Superdex™ 200 column can be used to purify the transcription reaction without a concentration step. In contrast, denaturing purification requires the development of refolding protocols that can be difficult for larger RNAs, whereas native purification takes advantage of folding during transcription.

## TPP riboswitch as a known RNA target

We attached the aptamer domain of the thiamine pyrophosphate (TPP) riboswitch[14] (86-nt) to our group II intron scaffold (*O.i.*-TPP). The TPP riboswitch binds to thiamine pyrophosphate and modulates gene expression, with a wide phylogenetic distribution in both prokaryotes and eukaryotes[14]. Multiple crystal structures of this riboswitch in the presence and absence of thiamine pyrophosphate revealed little difference in the overall RNA fold[27–29]. A crystal structure of a TPP riboswitch, as a crystallographic dimer, was recently solved in the absence of the ligand[30]. However, in this study, the density did not match the small-angle x-ray scattering reconstruction performed in solution, again emphasizing that crystal packing interferes with the sampling of conformations found natively in solution.

## High resolution reconstruction of the TPP riboswitch

High-resolution cryo-EM typically could not be applied to the TPP riboswitch, given its small size of 86 nucleotides (27.5 kDa). The TPP riboswitch was covalently attached to the solvent exposed domain III stem of the *O.i.* group IIC intron through a rigid helix (Fig. 3A). SHAPE-MaP[31] confirmed that the secondary structure of the 493-nt linked group II intron and riboswitch structure maintained the expected fold for both RNA domains (Figs. 3A and S2). SHAPE-MaP data also confirmed binding of the TPP ligand to the embedded riboswitch and revealed expected local conformational changes in the riboswitch[32]. These ligand-induced changes included modest rearrangement of the P2 and P3 helices, and reduced reactivities in the P3-L5 region and in the J3-2 and J2-4 elements. Cryo-EM data collection and subsequent processing were performed for the scaffold attached to the riboswitch (Fig. S3). Clear additional signal, corresponding to the attached riboswitch, was visible in all 2D class averages as compared to the 2D classes of the scaffold alone (Fig. 3B). Local refinement, focused on the scaffold, yielded a 3D reconstruction with a resolution of 2.4 Å for the core of the intron (Fig. 3C). A comparison of the resulting maps and models for the scaffold alone and with the TPP riboswitch embedded shows that the attachment process did not affect the overall fold of the scaffold (Fig. S4). We then performed local refinement on the riboswitch component, which yielded a reconstruction with a global resolution of 3.1 Å, and 2.5 Å for the ligand-binding pocket (Fig. 3D). The viewing distribution plot shows that embedding the riboswitch did not create a preferred orientation for the overall fusion construct (Fig. 3E). This resolution yielded clear density for the functional groups in this small molecule and enabled modeling of the TPP ligand with high confidence. Density for three magnesium ions, coordinated to the pyrophosphate moiety, are clearly visible (Fig. 4A, B). Only two metal ions were visualized binding to the pyrophosphate in previous crystal structures[28,29]. These magnesium ions are essential for binding of TPP to this riboswitch. The cryo-EM structure of this ligand-bound riboswitch exhibits the classic closed conformation observed in the crystal structures (Fig. S5).

In contrast, focused refinement of the ligand-free riboswitch revealed an open "Y" conformation at ~6 Å resolution (Fig. 5). In the ligand-free (apo) structure, the thiamine-sensing and pyrophosphate-sensing helices form a ~90° angle. In contrast, upon ligand binding, the RNA adopts a compact tertiary structure in which these two stems are parallel to each other and the P3 and L5 motifs form a long-range tertiary interaction, stabilizing the binding pocket for TPP. The apo state is also more dynamic due to the absence of the ligand and absence of stabilizing tertiary interactions. The thiamine sensing-stem adopts an alternate stable conformation in the apo state with two

major differences: the J2-4 element and the pyrophosphate-sensing stem are highly disordered, and the thiamine-sensing stem adopts a distinct conformation in which there is clear density for an additional helical groove that extends this helix compared to the bound state (Fig. 5A). These substantial conformational differences, visualized directly here, support a molecular model that rationalizes changes in the riboswitch structure as visualized by in-solution SHAPE probing of the ligand-free state (Fig. S2 and ref. 28)[33]. This apo structure is the same construct used for structure determination of the bound state, emphasizing that the ligand-free state is competent to bind TPP when scaffolded.

The dynamic nature of the unbound riboswitch explains why this state has not previously been captured using x-ray crystallography. The resolution of the apo state is limited due to the inherent dynamics of this destabilized RNA, yielding fewer particles going into the 3D reconstruction. We also observed fewer particles populating the holes compared to the bound state. We anticipate that collecting a significantly larger dataset would improve the resolution. Critically, the ligand-bound and apo structures support modeling the transition through which the riboswitch modulates gene expression upon ligand binding. In this model, the opening of the riboswitch allows it to interact with the downstream regulatory element and affect either translation or transcription through an alteration in RNA structure[12,13]. Our study provides the first cryo-EM data showing large-scale conformational changes in a riboswitch upon ligand binding, and which supports a specific mechanistic model for changes in gene expression.

## *RaiA* RNA as a target of unknown structure

The *raiA* RNA from *Clostridium acetobutylicum* was attached via its P1 helix to domain III of the group II intron scaffold. *RaiA* is larger than the TPP riboswitch and we initially collected a small dataset of only 1,328 movies, corresponding to ~3 h of data collection on a Titan Krios G3i cryo-EM microscope with a Gatan K3 detector (UC Berkley Cryo-EM Facility). We processed this small dataset, containing ~34,517 particles, to obtain a reconstruction with an overall resolution of 4 Å, with density approaching 3.5 Å in the core of the RNA (Fig. S7). This initial map showed clear separation of individual nucleobases and was of sufficient quality to determine the sequence register, which allowed us to build a preliminary model of the RNA structure de novo. We subsequently collected 16,183 movies, resulting in a reconstruction with an overall resolution of 3.0 Å, and 2.5 Å in the core (Fig. S7). This larger dataset improved the resolution and map quality to the point where we could clearly delineate the geometry of the backbone in regions with high distortion and assign accurate base geometries for the large number of non-canonical base pairs present in the *raiA* RNA.

## Overall structure of *raiA*

The map quality has allowed us to build a complete model for *raiA* and therefore define the overall architecture of this highly conserved bacterial ncRNA. Density is visible for every single nucleotide of the *raiA* structure. *RaiA* contains 8 conserved Watson-Crick helices (P1-P8) separated by internal loops (I) and junctions (J) (Fig. 6A). These internal loops and junctions consist of a large number of highly conserved non-canonical base pairs, which position the Watson-Crick helices (Fig. 6A). These conserved Watson-Crick and non-canonical structural motifs yield a highly compacted RNA (Fig. 6B) with two pseudoknots (pk-1 and pk-2) at its core (Fig. 7). The P3 stem loop (red) and J4-1c (cyan) contain the nucleotides that form one half of pk-1 and pk-2 respectively. The other halves of both pk-1 and pk-2 reside in the junction between P5 (purpleblue) and P6 (wheat), which is referred to as the pk-Loop (pk-L, green).

For its size, *raiA* is one of the most complex RNA structures observed to date. The P1 and P3 helices have an intricate geometry that

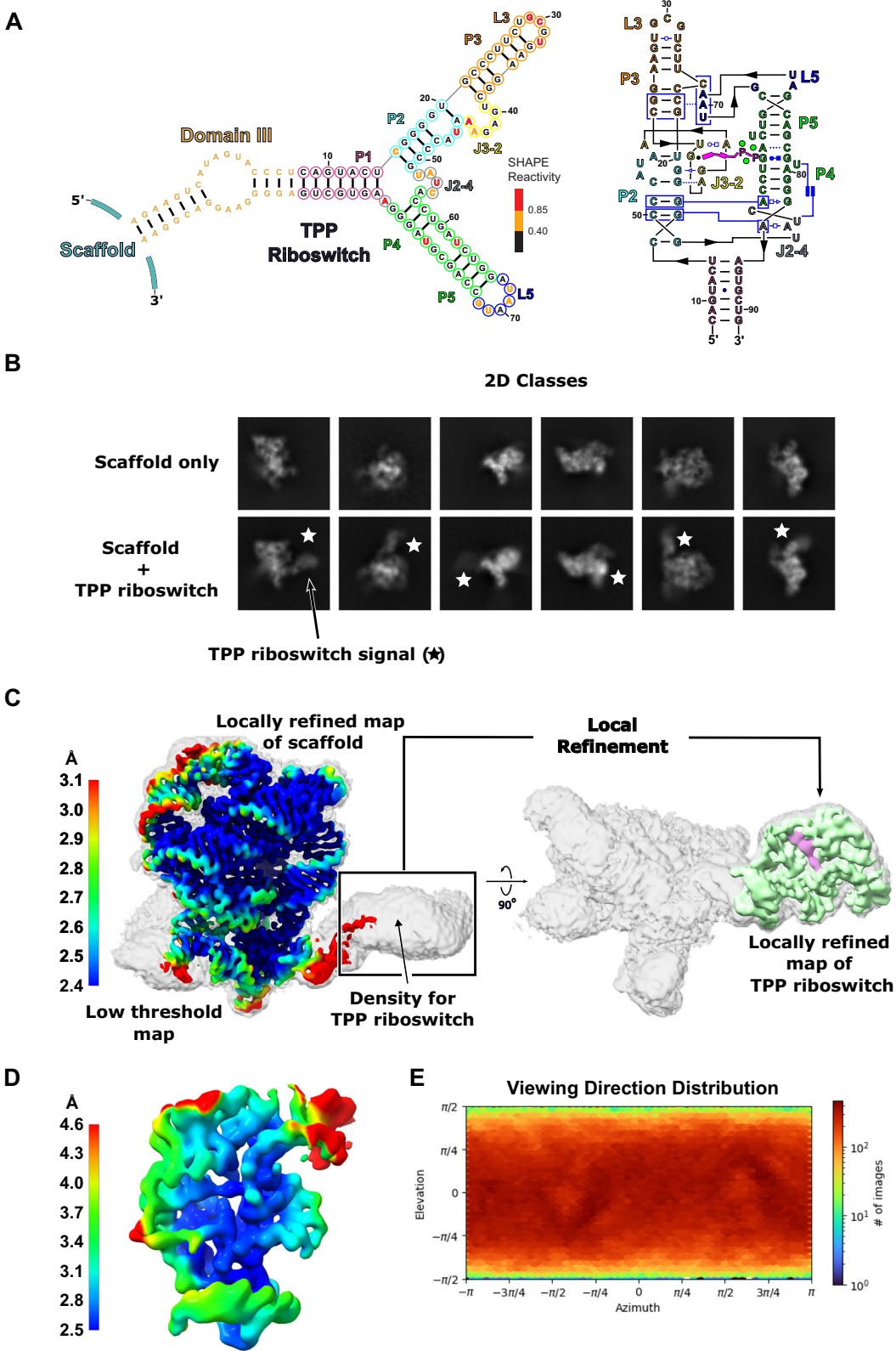

is organized by several internal loops and junctions that assist in positioning pk-1 in the core of *raiA*. The P1 stem terminates in a three-way junction formed by J1c-3a and J4-1c. This junction forms a discontinuous GNRA tetraloop (G20-A21-A22-A183) with a trans-U19:hA23 closing base pair that results in a bend of the helical axis of nearly 180 degrees (Figs. 8A and S8). This extremely tight turn over a relatively short sequence allows the *raiA* RNA to maintain a very compact

structure. The P1 and P3 stems then form a long-range tertiary interaction between J1b-1c and I3a-3b. The I3a-3b internal loop contains five adenosines (A28 and A43-A46) (Fig. S9) that organize the majority of this structural feature. A trans-H:W pair between A28:A45 forms in the middle of I3a-3b with A43 base stacking underneath this non-canonical base pair. This motif extrudes both A44 and A45 which interact directly with I1b-1c (Fig. S9). A trans-W:S pair forms between A44:G190 with

**Fig. 3 | High-resolution cryo-EM structure determination of the TPP riboswitch using the group IIC intron scaffold. A** Secondary structure of the TPP riboswitch attached to the scaffold through the stem of Domain III (gold) of *O.i.* Nucleotides in the TPP riboswitch aptamer domain are colored by SHAPE reactivity. Experiments were performed in the presence of ligand; for full SHAPE data, see Fig. S2. The riboswitch is also colored by helix for clarity. On the right, a secondary structure representation of the tertiary structure is shown. A cartoon depiction of thiamine pyrophosphate (magenta) is shown to emphasize the ligand binding pocket. The three magnesium ions observed in the cryo-EM data are shown as green spheres. The Leontis and Westhof nomenclature[54] is used to denote noncanonical base pairs. **B** 2D class averages of the scaffold alone and attached to the TPP riboswitch are shown. Signal corresponding to the TPP riboswitch (star) is clearly visible in the 2D classes. **C** A local resolution map of the scaffold (left) was generated from focused refinement and overlaid on a low threshold globally refined map (white). The locally refined map of the scaffold maintains high-resolution features after the TPP riboswitch is embedded. Density for the TPP riboswitch is visible in the globally refined map at low threshold. A locally refined map of the TPP riboswitch (green) is shown on the right and overlaid on the same low threshold globally refined map (white) for clarity. **D** A local resolution map for the TPP riboswitch is shown. The resolution surrounding the TPP binding site is 2.5 Å. **E** A viewing direction distribution plot is shown for the globally refined scaffold TPP data. The scaffold does not show any orientation preference with the added sequence of the riboswitch.

A46 and G191 base stacking on both sides of this conserved non-canonical base pair. This motif stabilizes I3a-3b in a geometry that creates an approximately 90° bend between the P3a and P3b helices (Figs. 8B and S9) ultimately pointing the P3 stem loop in the correct geometry to form pk-1 (Fig. 7A).

The P4 (cyan) and P5 stems form a single continuous helix through conserved non-canonical pairs at the base of P5 (Fig. 9A). These non-canonical pairs lead to the extrusion of A57 and A58 into the minor groove of pk-1 forming an A-minor motif (Fig. 9B, C). Immediately following P5 is the start of the pk-Loop which contains the first half of pk-2 and the second half of pk-1 (Fig. 6A). The pk-Loop then leads into the P6 stem, which consists of two variants. Variant 1 is a short stem loop and is found in 32% of the identified *raiA* sequences while variant 2 consists of a larger multi-stem domain (P6, P7, and P8) (Fig. 10A) in the remaining 68% of sequences. The *raiA* motif from Clostridium acetobutylicum used in this study belongs to the variant 2 class and contains a conserved E-Loop (Fig. 6A) in P8 that is found in 20% of identified sequences. The P8 stem loop in this construct contains 19 base pairs with 13 being non-Watson/Crick and extends parallel to the P4/P5 stem loop. The most highly conserved region within P8 is the E-Loop motif, which further stabilizes pk-1 in the core of the RNA through a ribose zipper interaction (Fig. 10B)

Finally, pk-2 is formed between the pk-Loop and J4-1c. Two inner shell coordinated magnesium ions are observed stabilizing the nucleotides C80 and U82 that contribute the first half of pk-2 (Fig. 7B). Nucleotides G179 and A184 form a trans-S:W base pair that creates a tetraloop containing the second half of pk-2 (C181/A182). The pk-1 and pk-2 pseudoknots are separated by a single nucleotide (G84) which base stacks with the first nucleotide (A33) of the P3 loop (Fig. 7C). Taken together, *raiA* forms a compact structure through short-range and long-range interactions between highly conserved nucleotides. SHAPE-MaPreveals low SHAPE reactivities across the majority of the *raiA* RNA sequence indicating that most nucleotides are highly constrained (Fig. S10). A lack of nucleotide flexibility, even in single-stranded regions, supports the structural stability observed in the cryo-EM data.

### Cryo-EM Data Processing of Scaffolded RNAs

The processing workflow for scaffolded RNAs requires special considerations. As a first step, we refined the entire assembly using CryoSPARC[34] to assess the quality of the cryo-EM data. At this stage, the scaffold serves as an internal control and should have a resolution of ~2.5 to 3 Å. The overall fold of the group II intron should be maintained and density emanating from domain III for the target should be visible at a low map threshold. The signal corresponding to the scaffold is then subtracted from the particle images. In the case of smaller targets, such as the 86-nt TPP riboswitch, focused refinement is performed following signal subtraction. A focused refinement strategy compensates for the lower signal from the low-mass riboswitch, maintains alignment information from the initial global refinement, and results in a higher resolution reconstruction. In contrast, larger RNAs, like the 210-nt *raiA* molecule, do not require focused refinement because there is sufficient signal for subsequent non-uniform refinement.

## Discussion

We have developed a technology for high-resolution structure determination of small RNAs, through fusion to a group II intron scaffold. This scaffold strategy is the first to demonstrate cryo-EM structure determination of target RNAs to better than 3 Å resolution, and yields two landmark results. For the TPP riboswitch, the resolution observed here is the highest ever achieved by cryo-EM for an RNA less than 100 nucleotides. For the *raiA* motif, our scaffolding study was characterized by rapid progress from initial selection of the RNA target to determining its structure at sub-3 Å resolution over a period of ~3 weeks.

The TPP riboswitch undergoes large-scale conformational changes when comparing the ligand-free to the bound state. The ligand-free state of the riboswitch reveals an initial Y-shaped fold that, when bound to ligand, closes into a compact fold. In addition to this long-range movement, the thiamine sensing stem in the ligand-free state has a more elongated helix with the appearance of an additional helical groove (Fig. 5A) compared with the bound state. This is consistent with the rearrangement of the secondary structure in the base of this stem to form additional pairing interactions, which agrees with SHAPE probing data (Fig. S2C). It is likely that a rearrangement in secondary structure is required for the interaction with the pyrophosphate group upon ligand binding.

The large conformational difference between the bound and apo forms supports a mechanism for riboswitch function, consistent with previous hypotheses[12–14]. The native TPP riboswitch features a downstream regulatory element consisting of a stem loop containing the start codon of the mRNA[14]. In this model, the ligand-free riboswitch initially exists in a relaxed, dynamic Y-shaped form and interacts with the regulatory stem loop to render the start codon accessible for translation initiation[14] (Fig. 5B). Once the riboswitch binds the ligand, the riboswitch clamps down and forms a compact structure that disengages the riboswitch from the regulatory stem loop, creating new structures that sequester the start codon and inhibit translation. Large conformational changes, of similar magnitude upon ligand binding, are likely a general mechanism for riboswitch function, but have been difficult to visualize directly.

The core of the *raiA* RNA was visualized at a resolution of 2.6 Å, which allows de novo modeling of the RNA structure and distinguishing between purine and pyrimidine residues. It was previously hypothesized that *raiA* is a ribozyme and not a riboswitch due to the lack of a downstream expression platform[22]. The highly ordered nature of the *raiA* structure suggests that it is possible that this could be a novel class of structured RNA, including functioning as a ribozyme. *RaiA* is highly expressed and a dominant cellular RNA, suggesting it plays an important role in bacteria[22]. CRISPR-based knockouts of the entire *raiA* RNA gene revealed its importance in sporulation and biofilm formation[22].

 

The high resolution structure of *raiA* RNA now allows detailed CRISPR-based mutagenesis of individual nucleotides to assess the structure with in vivo biological function. This approach flips the typical paradigm in RNA biochemistry, which usually involves extensive biochemical characterization before attempting structure determination. High resolution structure determination can now be carried out as part of the initial characterization of newly discovered RNA motifs. Knowledge of the structure may also

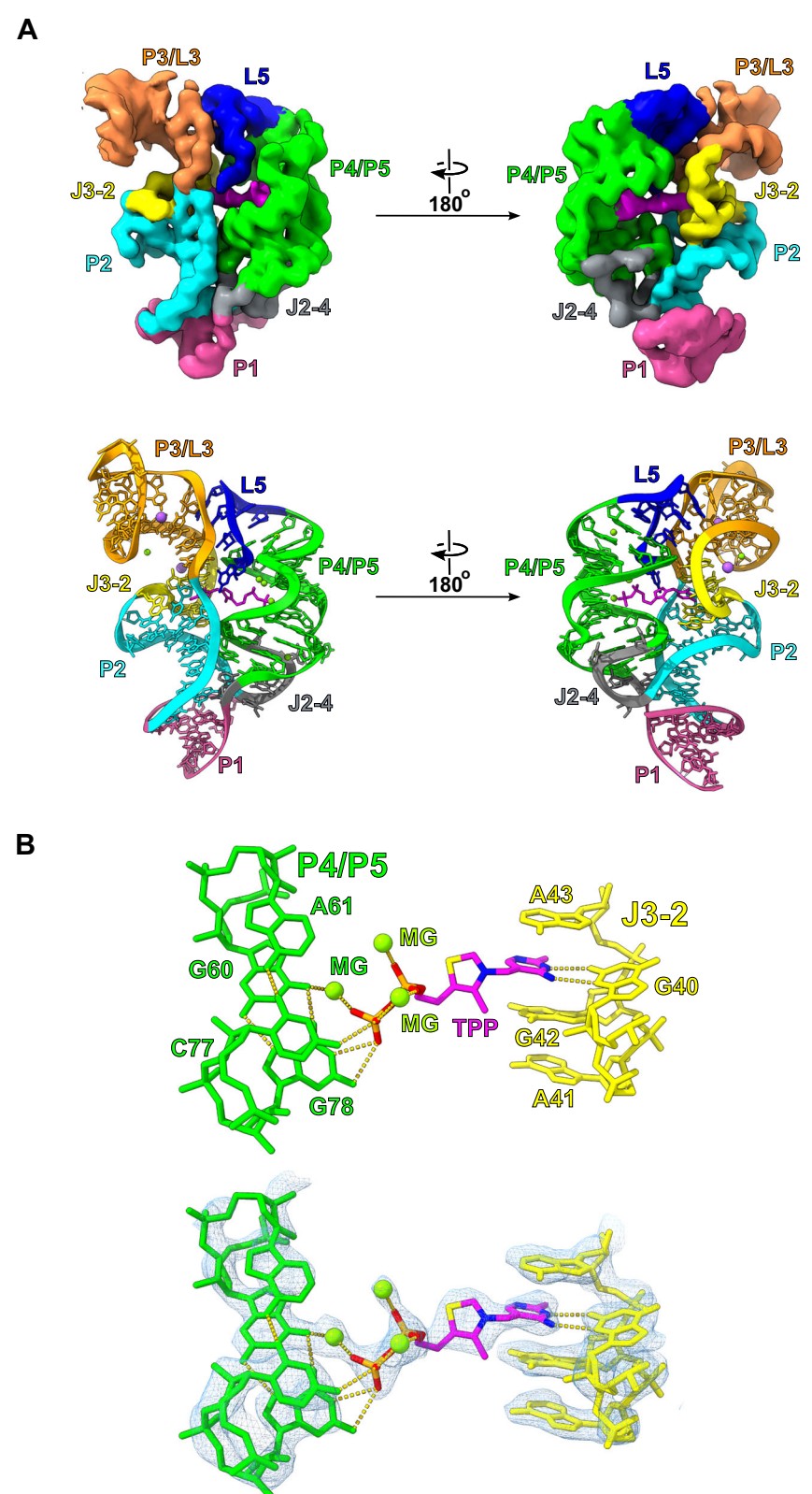

**Fig. 4 | High-resolution cryo-EM structure of the TPP riboswitch. A** The unsharpened cryo-EM map as well as the corresponding model are shown for the scaffolded TPP riboswitch. The density and model is colored to match the secondary structure in Fig. 3A. **B** The high-resolution density surrounding the thiamine pyrophosphate binding site allowed for the precise characterization of ligand binding by the riboswitch. The model overlaid with density is shown below.

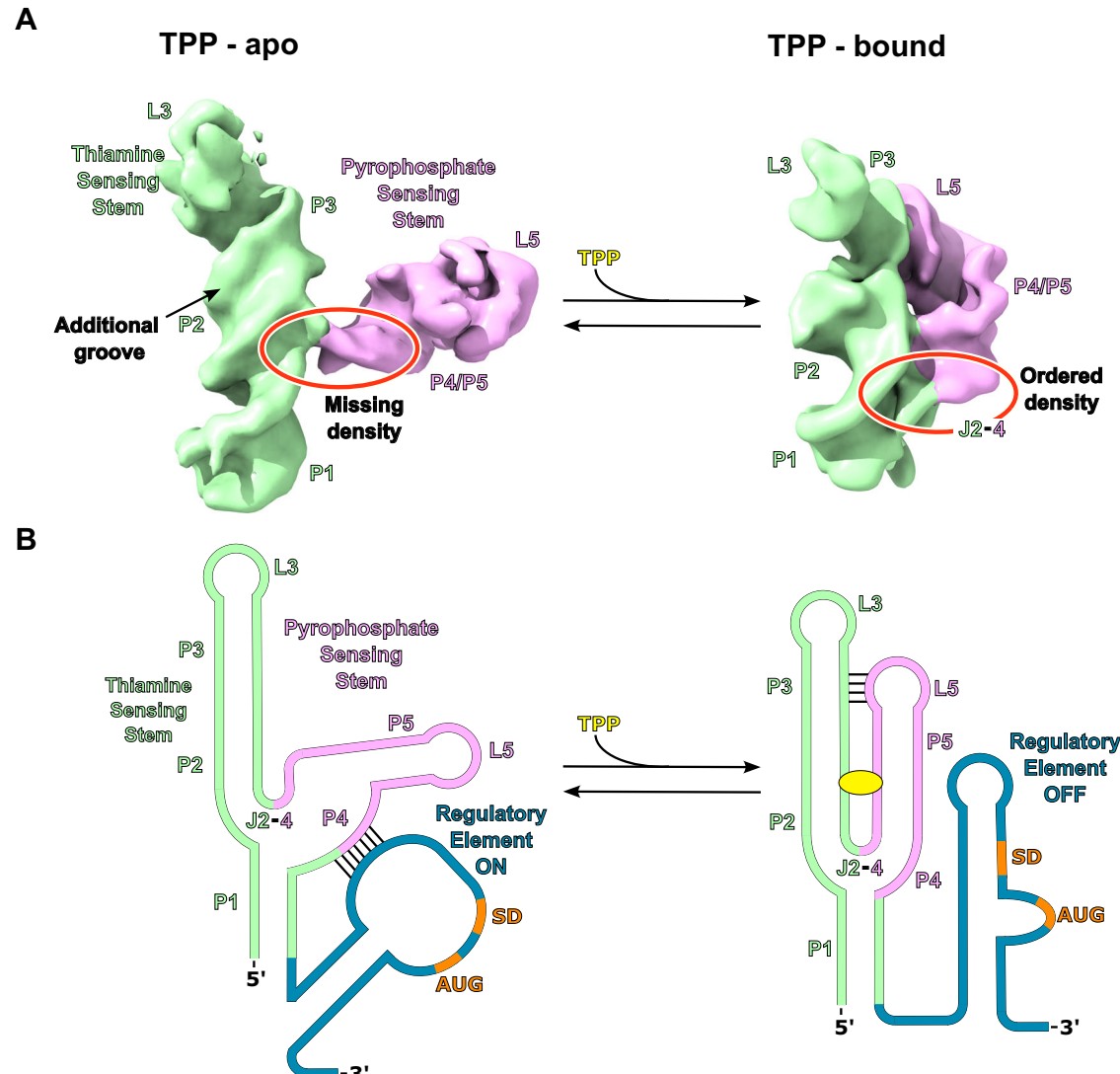

**Fig. 5 | A model for translational regulation by the TPP riboswitch. A** Density for the apo and TPP bound states of the TPP riboswitch are shown. The TPP bound map was low-pass filtered to 6 Å. The P3-L5 interaction is disengaged in the apo state (left) allowing the pyrophosphate sensing stem (purple) to rotate, causing the riboswitch to adopt an overall Y shaped conformation. In the apo state the base of the L5 stem loop is largely disordered with portions of P1, J2-4, and P4 missing density (circled red). In contrast, the thiamine sensing stem (green) appears to have an alternate stable conformation in the apo state with the appearance of an additional minor groove. When TPP is bound (right), the P3-L5 interaction is engaged, and the riboswitch compacts leading to ordered P1, J2-4, and P4 regions (circled red) forming a compact three-way junction. **B** The above structural information provides insights into the mechanism of translational regulation by the TPP riboswitch. In the apo state, the base of the L5 stem loop is disordered allowing it to interact with the downstream regulatory element (blue) through complementary base pairing. This interaction causes a rearrangement of the regulatory element structure making the Shine-Dalgarno sequence (SD, orange) accessible for the ribosome to initiate translation. The stable conformation of the thiamine sensing stem in the apo state likely promotes the formation of this L5 stem loop/regulatory element interaction. In the bound state, the equilibrium is shifted to the compacted state with an ordered L5 stem loop, which prohibits the interaction of the L5 stem loop with the regulatory element. This ultimately leads to a loss in ribosome initiation and translation.

allow for the detection of additional *raiA*-like motifs in the genomes of many more bacterial species, beyond the 2500 seen to date.

Based on its importance in biofilm and spore formation, *raiA* may represent an antibiotic target. Given its high degree of conservation across many bacterial species, *raiA* appears to be a better drug target than most riboswitches. We have identified multiple tertiary contacts and pockets that might serve as ligand binding sites to possibly disrupt its function in vivo. Our scaffolding strategy enabled structure determination of *raiA* from only 1200 micrographs, supporting efficient screening of both drug targets and ligands.

This work shows that it is possible to obtain high resolution structures of small RNAs and novel uncharacterized ncRNAs in the 86 to 210 nucleotide size range using cryo-EM. There should be no lower limit to RNA size using this approach. The likely upper limit will occur when an attached RNA target overwhelms the favorable properties of the group II intron scaffold. The ability to visualize routinely a small molecule bound to RNAs with complex structures[35] using cryo-EM now creates a broad opportunity to use this approach in drug discovery efforts targeting RNA structures.

## Methods
### Plasmid cloning
The mutant *O.i, O.i.*-TPP, and *T.el*4h-*O.i.* genes were synthesized (Genscript) and cloned into a pUC57 vector using the EcoRV restriction site. The cloned plasmids were transformed into DH5α cells. The *O.i.*

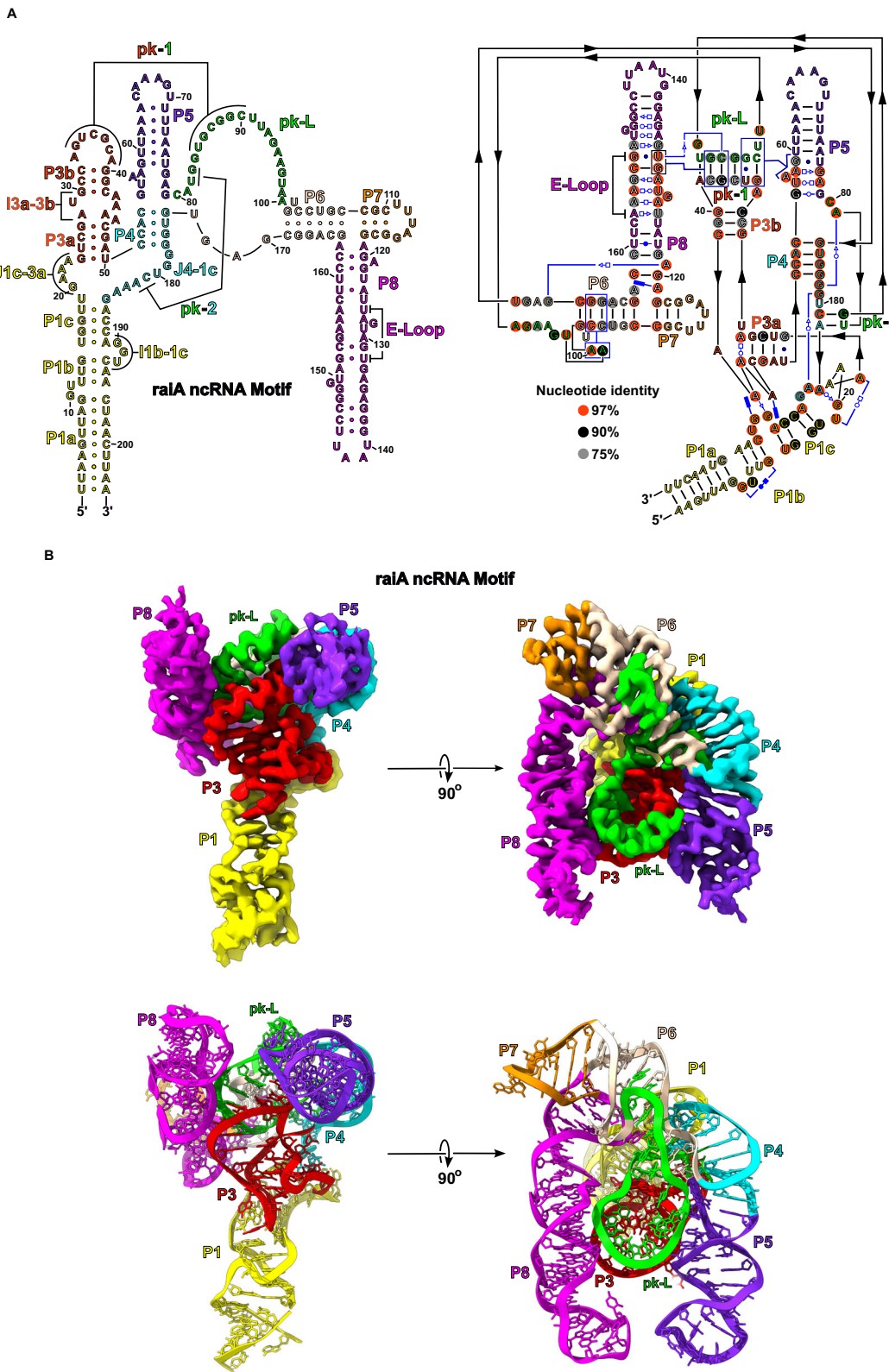

**Fig. 6 | cryo-EM structure of the *raiA* ncRNA motif. A** The secondary structure of the *raiA* motif from *Clostridium acetobutylicum* is shown. The RNA consists of seven helical stems (P1-P8). Each individual helix is colored for clarity and relevant tertiary interactions are labeled. The pk-Loop (green) is made up of the junction nucleotides between P5 and P6. On the left, a secondary structure that more closely follows the tertiary structure is shown outlining the organization of the helices relative to each other in three-dimensional space. Symbols to denote noncanonical base pairs are shown in blue and use the Leontis and Westhof nomenclature[54]. **B** The unsharpened cryo-EM map and corresponding model of the scaffolded *raiA* non-coding RNA motif is shown. Density has been colored to match the secondary structure.

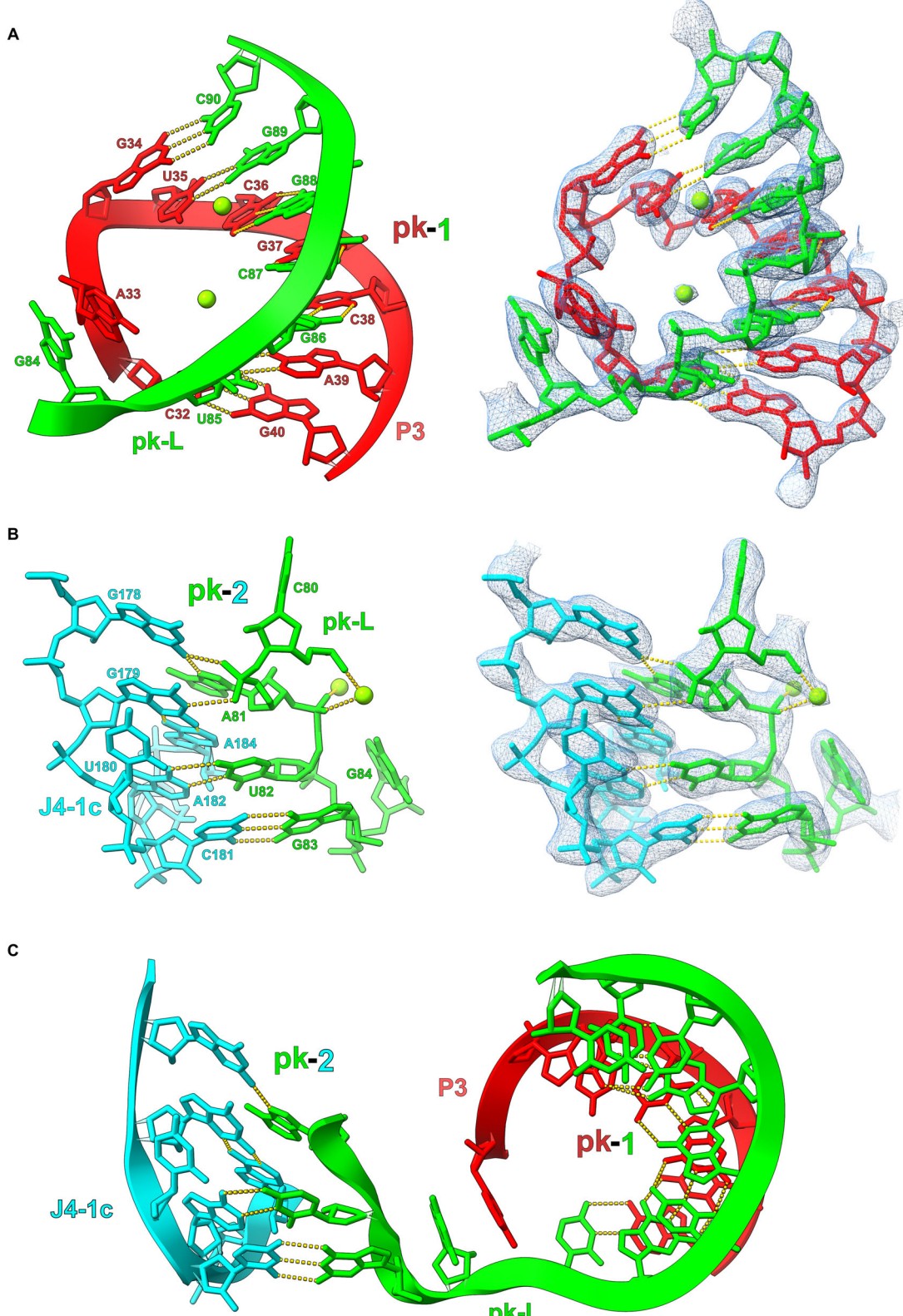

**Fig. 7 | The pk-1 and pk-2 interactions form the core of the *raiA* motif. A** The P3 loop (red) of *raiA* forms an extended pseudoknot with the pk-Loop (green) (pk-1). The pseudoknot consists of 7 consecutive cis-W:W base pairs. **B** pk-2 forms between the pk-Loop (green) and the junction between P4 and P1c (J4-1c) (cyan).

This pseudoknot is not as extensive as pk-1 and forms 2 cis-W:W base pairs. **C** The sequence of pk-1 and pk-2 contained within the pk-Loop is only separated by a single nucleotide (G84). G84 base stacks with first nucleotide of the P3 Loop (A33) to further stabilize the pk-1/pk-2 structure.

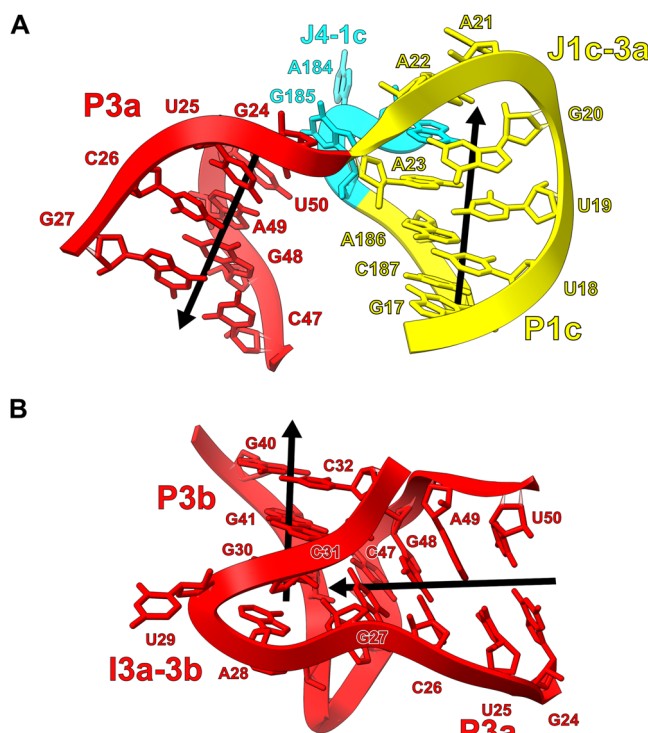

**Fig. 8 | The P1 and P3 stems contain significant bends to their helical axis. A** The J1c-3a and J4-1c junctions turn the helical axis of P1 relative to P3 by ~180°. The helical axis of the P1 and P3 stems is represented by black arrows. **B** The I3a-3b internal loop forms a ~90° bend of the helical axis between the P3a and P3b stems.

and *O.i.*-TPP genes contained a 4-nt 5′ exon and a 3-nt 3′ exon followed by a BamHI cut site. The *T.*el4h-*O.i.* gene contained a 19-nt 5′ exon and a 9-nt 3′ exon followed by a BamHI cut site. The *T.el.*4h maturase expression plasmid used in this study was previously described[10]. The *O.i.-raiA* gene was synthesized by Genewiz using their PriorityGENE service and inserted into their pUC-GW-Amp vector. The *O.i.-raiA* gene contained a 4-nt 5′ exon and a 3-nt 3′ exon followed by a BamHI cut site. The scaffold of the *O.i.-raiA* gene was modified from the one used in the TPP study. The following modifications were made: nucleotides 349 and 390-412 were deleted, the GAAA tetraloop (275-278) was mutated to a UUCG, U347 was mutated to an A, and A348 was mutated to a U. Use Figs. 2A and S6 to reference nucleotide numbering.

### In vitro RNA transcription and purification
The *O.i, O.i-*TPP, *T.el*4h-*O.i.*, and *O.i-raiA* plasmids were linearized using an engineered BamHI restriction site (NEB). 50 µg of template DNA was added to a total volume of 1 mL of in vitro transcription buffer (50 mM Tris-HCl pH 7.5, 25 mM MgCl$_2$, 5 mM DTT, 2 mM spermidine, 0.05% Triton X-100, and 5 mM of each NTP). T7 polymerase and thermophilic inorganic phosphatase was added to begin RNA synthesis. The reaction mixture was incubated at 37 °C for 3 hrs. CaCl$_2$ was added to a final concentration of 1.2 mM along with Turbo DNase and placed at 37 °C for 1 hour to fully digest the DNA template. Proteinase K was subsequently added and incubated at 37 °C for an additional hour. The resulting solution was centrifuged to remove any precipitate and then filtered through a 0.2 µm filter. The filtered solution was buffer exchanged a total of 7 times, each time using 14 mL of filtration buffer (5 mM Na-cacodylate pH 6.5 and 4 mM MgCl$_2$) and a 100 kDa molecular weight cut-off filter. After the final buffer exchange step, the RNA was concentrated to approximately 10 mg/mL for use in downstream cryo-EM experiments.

### SHAPE-MaP RNA structure probing
SHAPE-MaP assays were performed essentially as described[36]. Briefly, 10 pmol of either the TPP riboswitch or *raiA* RNA was diluted in 19 µL probing buffer (300 mM HEPES, pH 8.0, 2 mM MgCl$_2$, 40 mM NaCl). In the case of the ligand-bound TPP riboswitch, 10 pmol RNA was mixed with 1 uL of TPP (final concentration, 50 µM). After incubation for 15 min at 37 °C, 9 uL of RNA-ligand solution was added to 1 uL 1 M 2A3[37] and mixed thoroughly. After incubation for 20 min, the reaction was quenched by addition of 5 uL of 1 M DTT. No-ligand reactions were conducted in parallel. RNA was purified (G-25 spin column) and subjected to reverse transcription (SuperScript II, Invitrogen) using a heat-cool protocol [25 °C for 10 min, 42 °C for 90 min, 10 cycles of (50 °C for 2 min, 42 °C for 2 min), 70 °C for 10 min]. DNA libraries were prepared from cDNA using a two-step PCR process (Q5 Hot Start High-Fidelity DNA Polymerase, New England Biolabs). PCR 1 used Step-1 Forward primer and Step-1 Reverse primer, and performed for 20 cycles; PCR 2 used Universal Forward primer and Universal Reverse primer, and performed for 10 cycles. All cDNA or dsDNA were purified (Mag-Bind TotalPure NGS beads, Omega), quantified (Qubit dsDNA Quantification Assay Kits, Life Technology), and visualized to confirm integrity (2100 Bioanalyzer Instrument, Agilent). The libraries were sequenced on a MiSeq system (Illumina). Mutations were aligned and parsed using ShapeMapper (v2.15)[38] using default parameters.

TPP RT primer sequence: 5′-CTAATAGAGT AGAGCGAACT CCTCTC-3′.

TPP Step-1 Forward primer: 5′-CCCTACACGA CGCTCTTCCG ATCTNNNNNT TATGTGTGCC CGGCATG-3′.

TPP Step-1 Reverse primer: 5′-GACTGGAGTT CAGACGTGTG CTCTTCCGAT CTNNNNNCTA ATAGAGTAGA GCGAACTCCT-3′.

*raiA* RT primer: 5′-ACACCACCGTACGTACC-3′. *raiA* Step-1 Forward primer: 5′-GACTGGAGTTCAGACGTGTGCTCTTCCGATCTNNN NNCTTTCGAGCTCAGAAGTCAG -3′. *raiA* Step-1 Reverse primer: 5′-CCCTACACGACGCTCTTCCGATCTNNNNN GTTCAAGCATGGCGC-3′.

Universal Forward primer: 5′-AATGATACGG CGACCACCGA GATCTACACT CTTTCCCTAC ACGACGCTCT TCCG-3′. Universal Reverse primer: 5′-CAAGCAGAAG ACGGCATACG AGAT (8nt-barcode) GTGACTGGAG TTCAGAC-3′.

### Group IIB intron RNP assembly
The RNP used in this study was assembled and purified as previously described[10].

### EM sample preparation
For all grids prepared with TPP bound, 100 uL of 2 mg/mL RNA was incubated with 1 mM thiamine pyrophosphate at room temperature for 15 minutes prior to freezing. For all grids prepared with the riboswitch in the apo state as well as the *O.i.-raiA* motif, this binding step was omitted. To freeze the grids, 3.5 µL of freshly prepared RNA sample at 2.0 mg/mL was applied to a glow discharged (40 mBar, 15 mA for 30 s using a PELCO easiGlow) copper R1.2/1.3 300-mesh grid (Quantifoil). The grid was blotted with a filter paper (Whatman No.1) at 4 °C in a cold room before plunging frozen into liquid ethane/propane (37.5/62.5) mix using a manual plunger. For the *T.el*4h-*O.i* RNP specimen, 3.5 µL of freshly prepared RNP at 1 mg/mL was applied to a glow discharged (40 mBar, 15 mA for 120 s using a PELCO easiGlow) UltrAuFoil R1.2/1.3 300-mesh grid (Quantifoil). The grid was blotted with a filter paper (Whatman No.1) at 4 °C in a cold room before plunging frozen into liquid ethane/propane (37.5/62.5) mix using a manual plunger.

### Cryo-EM data collection and processing
Movies were collected on the same Titan Krios microscope (Thermo Fisher) operating at 300 keV, equipped with a K3 Gatan direct electron detector, and were collected with a magnification of 105,000x for a physical pixel size of 0.811 Angstroms and camera operating in super-resolution mode (0.4055 Å/pixel). *O.i-raiA* movies were collected with

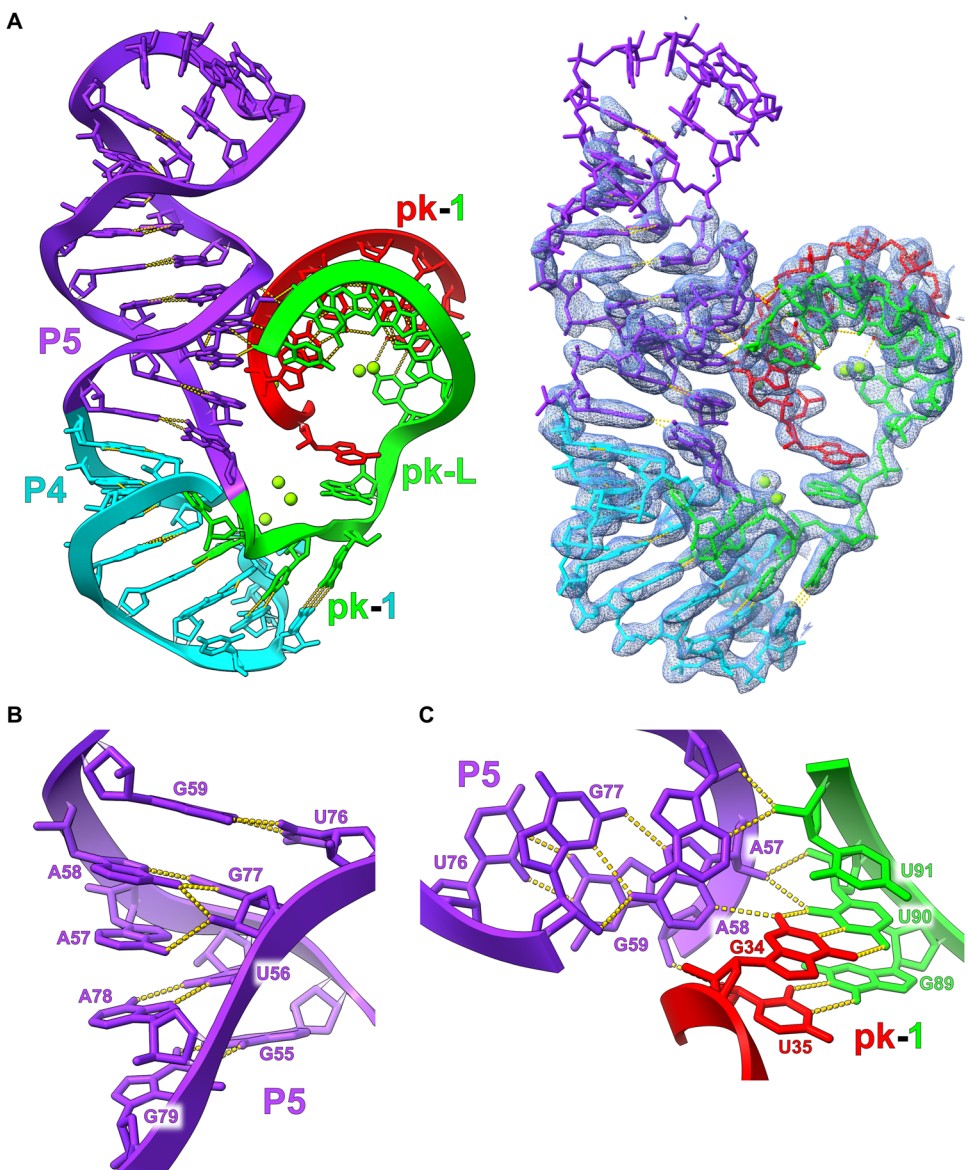

**Fig. 9 | P4 and P5 form a continuous helix and stabilize pk-1 through an A-minor motif. A** The structural organization of the P4 and P5 helices is shown relative to pk-1. The minor grooves of P5 and pk-1 point towards each other and form an extended interface of hydrogen bonds. **B** The base of P5 consists of several noncanonical base pairs that forms a continuous helix between P4 and P5, and extrudes A57 and A58 from the helical axis. **C** A57 and A58 from the base of P5 extend into the minor groove of pk-1 forming an extended A-minor motif interaction.

a magnification of 105,000x for a physical pixel size of 0.848 Å and camera operating in super-resolution mode (0.424 Angstroms/pixel). Movies collected for *O.i.*, *O.i.*-TPP, *O.i.*-TPP-Apo and *O.i.*-*raiA* were collected as 50 dose-fractionated frames with a dose rate of 8 e-/pixel/s for a total exposure time of 4.5 s and total dose of ~50 e-/Å². The *O.i.* and *O.i.*-TPP, *O.i.*-TPP-Apo datasets were collected with a defocus range of −0.5 μm to −1.5 μm, whereas the *O.i.*-*raiA* dataset was collected with a defocus range of -0.8 to -2.0 um. The *T.el.*-*O.i.* dataset was collected with the same parameters described previously (Ref) as reported in table S1, with the exception of a defocus range of −0.8 μm to −1.9 μm, and an exposure time of 5.4 s. All movies were semi-automatically collected using SerialEM[39]. 5724 (*O.i.*), 30,314 (*O.i.*-TPP), 15,876 (*O.i.*-TPP-Apo), 2596 (*T.el.*-*O.i.*), 1328 (*O.i.*-*raiA* Small), and 16,183 (*O.i.*-*raiA*) movies were collected with these parameters.

For the *O.i.* dataset, 5,724 movies were collected and imported into Relion 3.1[40–42]. Super resolution movies were binned by two and motion-corrected using MotionCor2[43]. These motion-corrected

micrographs were imported into CryoSPARC 3.3.2[34], and CTF was estimated using Patch CTF. Micrographs were visually inspected and 343 micrographs were thrown out due to poor particle distribution and ice contamination. Particles were picked on the remaining 5,381 micrographs using a general model from Topaz 0.2.5[44] as implemented in CryoSPARC. These ~1.6 million particles were extracted with a box size of 288 pixels downsampled to 64 pixels, for a final pixel size of 3.6 Å/pixel. Particles were ran through 2D classification, and a small subset of particles was selected for ab initio model building. This model was then used for four subsequent rounds of 2-class hetero-geneous refinement to classify the ~1.6 million particles into a 3D class that looked like the group II intron structure. The resulting 725,736 particles were extracted as 320 pixel boxed particles downsampled to 160 pixels for a pixel size of 1.6 Å/pixel. The particles were then sub-jected to several additional rounds of 3D classification, refinement, and CTF refinement, before the final set of 607,776 particles were extrac-ted with a box size of 360 pixels downsampled to 240 pixels, for a final

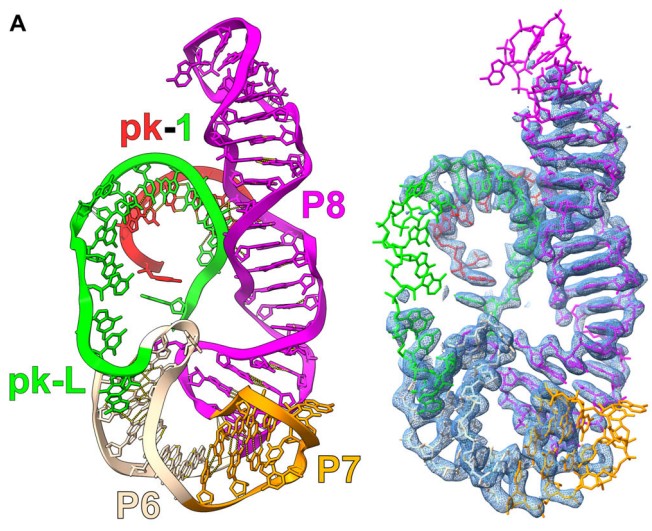

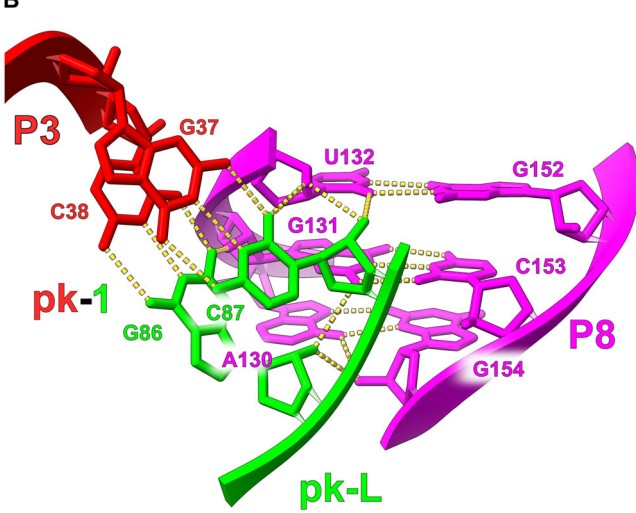

**Fig. 10 | The P8 stem loop also interacts with pk-1 through an A-minor motif. (A)** The organization of the P6, P7, and P8 stems are shown relative to pk-1. P8 consists of numerous noncanonical base pairs and, as a result, exhibits a distorted helical geometry. The minor grooved of P8 and pk-1 can be observed interacting. **(B)** The conserved E-Loop region of P8 forms through multiple consecutive non-canonical base pairs. This architecture results in a helical geometry capable of forming an A-minor motif with the opposite side of pk-1 relative to the P4/P5 stem loop.

dataset was used for picking ~633,000 particles on the full 1,775 micrographs, 502,500 of which were then extracted at a box size of 512 pixels downsampled to 128 pixels, for a final pixel size of 3.2 Å/pixel. The 3D volume generated by the initial refinement was used as the starting template for three rounds of consecutive 2-class heterogeneous refinement, and the resulting 331,439 particles were put through refinement and extracted with a box size of 512 pixels (0.811 Å/pixel). The 322,683 particles were subjected to duplicate removal (20 Angstrom separation), another round of 2-class heterogeneous refinement, refinement, CTF refinement, and 2D classification, for a final set of ~283,000 particles. The GSFSC (0.143) global resolution of this particle set after non-uniform refinement was 2.88 Å as reported by cryoSPARC. These particles were then subjected to hetero-refinement and 3D classification to select particles where the embedded *O.i.* insert was fully intact. The 34,822 particles selected from classification were used for masked local refinement around the *O.i.* insert, which had a GSFSC resolution of 4.02 Å as reported in CryoSPARC.

For the *O.i.*-TPP-Apo dataset, 15876 super resolution movies were collected and imported into cryoSPARC 4.2.1. Movies were binned by two, and motion-corrected and CTF estimated using patch motion correction and PatchCTF, respectively. Exposures with CTF resolution fits above 10 Å, relatively thick ice, and high values for the predicted stage tilt angle, were removed, leaving 11,485 micrographs for further processing. Due to difficulties with initial blob picking, a 1,800 micrograph subset was used for template picking using an *O.i.* map processed from a previous dataset. These ~800,000 particles were used for two rounds of 3-class hetero-refinement, with the final 93,000 particles used to generate a new template for particle picking against the full ~11,000 micrograph dataset. The ~8,000,000 particles picked were extracted with a 336 pixel box downsampled to 84 pixels (3.2 Å/pixel). These particles were subjected to four consecutive rounds of hetero-refinement (4 classes, 2-class, 2-class, and 2-class). The resulting 1,212,955 particles were then refined and re-extracted to a box of 128 pixels (2.3 Å/pixel). Duplicate particles (50 angstrom radius) were removed, and remaining particles were used for a 5-class hetero-refinement. The 1,056,211 particles selected were refined, and extracted with a 256 pixel box (1.27 Å/pixel), and then put through another round of refinement and CTF refinement. These 1,032,440 particles were then ran through a 10-class hetero-refinement, and 2 classes that corresponded to the highest resolution densities were selected. These ~600,000 particles were then subjected to multiple rounds of refinement and CTF refinement to improve resolution, with one final 2D classification job used to remove particles with poor 2D classes. These 547,101 particles were re-extracted with a box size of 448 pixels downsampled to 288 pixels (1.26 Å/pixels) and non-uniform refined to a final global GSFSC resolution of 2.78 Å as reported by cryoSPARC.

For the apo-TPP embedded structure, those ~547,000 particles were put through multiple rounds of 3D classification and hetero-refinement to remove conformations for the P2 and P3 stems that did not have resolvable densities, or corresponded to conformations that were not relevant to the TPP riboswitch. Classes that had intact density for an open conformation were pooled together, for a final particle set of 106,285 particles. In parallel, a mask around the *O.i.* scaffold was generated for local refinement of the scaffold, and these 3D coordinates were used for signal subtraction of the scaffold from the particle images. Then, the 3D coordinates of the ~106,000 particles after non-uniform refinement were used as a starting point for local refinement of the apo-TPP riboswitch in its open conformation. Gaussian priors of an 8 degree rotation and 4 Å shift were used for masked local refinement of the apo state, which had a final GSFSC of 4.84 Å as reported in CryoSPARC.

The basic workflow for the high resolution reconstruction of the TPP-riboswitch is outlined in figure S3. In brief, three separate data collections of the *O.i.*-TPP construct were collected (7763 movies, 6303

pixel size of 1.22 Å/pixel. These ~600,000 particles were put through a single round of 3D classification with alignment (562,162 particles selected), 2D classification (458,691 particles selected), and 9-class heterogeneous refinement (382,753 particles selected) for non-uniform refinement[45]. This map refined to a final GSFSC resolution of 2.62 Å as reported in cryoSPARC.

For the *T.el.*-*O.i.* dataset, 2596 movies were collected at a 30° stage tilt and imported into cryoSPARC 4.2.1. Super resolution movies were binned by two and motion-corrected and CTF was estimated using Patch Motion Correction and PatchCTF, respectively. 1775 motion corrected micrographs were selected for further processing with good particle distribution and little ice contamination. 100 micrographs were randomly selected for initial particle picking using blob picker, and these ~60,000 particles were extracted for 2D classification. Good 2D references corresponding to ~22,000 particles were selected for ab initio model building and refinement. The 3D reference from this

movies, 16248 movies) with identical parameters, and pooled together as 30,314 super resolution movies. These movies were initially imported into cryoSPARC 4.2.1 for validation and a first reconstruction, and used a workflow similar to that implemented for the *O.i.*-TPP-apo construct. The first maps from cryoSPARC had a resolution close to 3.7 Å/pix, and so the super resolution movies were imported into Relion 4.0 as separate exposure groups, and put through a standard workflow of particle picking, 3D classification, refinement, and particle polishing at the end. These particles were then exported into cryoSPARC, where they underwent the process of resolution improvement through non-uniform refinement and particle defocus refinement for a final GSFSC resolution of 2.53 Å. Masked refinement of the scaffold was then used for signal subtraction from the polished particles, and the signal subtracted particles were recentered on the density for the TPP riboswitch. This TPP riboswitch density was locally refined with a Gaussian prior of a 3° rotation and 3 Å shift, with a final reported GSFSC resolution of 2.96 Å in CryoSPARC.

The small *O.i.-raiA* dataset (1328 movies) was imported into cryoSPARC 4.5.1 using cryoSPARC Live, where super resolution movies were binned to 0.848 Å/pixel and CTF was estimated using Patch Motion Correction and Patch CTF, respectively. 118 exposures were removed due to low resolution CTF fits, and 200,662 particles were then picked on these 1210 micrographs using a Topaz 0.2.5 general model. 171,146 particles were then extracted with a 108 pixel box (3.39 Å/pixel) and subjected to a single round of 2D classification, ab initio model generation, and heterogeneous refinement (4 classes), with 34,293 particles being the final result. 33,096 particles were extracted (256 pixels 1.70 Å/pixel) with a GSFSC resolution of 3.63 Å after non-uniform refinement. A mask was then generated around the *raiA* insert, and this region of density had a final resolution of approximately ~4.7 Å. 2507 particles from 47 exposures were used to train a new Topaz model to pick *O.i.-raiA* particles on these micrographs.

Using the Topaz model, 263,878 particles were picked and then extracted from 1,194 micrographs (108 pixels, 3.4 Å/pixel). These particles were subjected to a single round of 2D classification to remove junk particles, and then two rounds of heterogeneous refinement (6 classes) to pick the highest resolution classes to pool together. These particles were refined, re-extracted (324 pixels, 1.34 Å/pixel), and then subjected to non-uniform refinement with CTF defocus correction, for a final GSFSC resolution of 3.4 Å. After attempting particle subtraction, heterogeneous refinement, 2D classification, 3D classification, and local refinement, the resolution and reconstruction of the *raiA* insert did not improve. After 2D classification to remove junk particles, 82,698 particles were re-centered on the *raiA* insert and re-extracted (192 pixels, 0.848 Å/pixel). A homogenous reconstruction of the *raiA* density was then used as a reference for a single round of heterogeneous refinement (4-class), and density corresponding to an intact reconstruction of the *raiA* insert was used as selection criteria for a single 34,517 particle class, with the other three classes being representative of the *O.i.* scaffold or poor realignment. The particle sets utility was used to select these 34,517 particles from the original 324 pixel downsampled particles, and the *raiA* insert density was locally refined with a Gaussian prior of a 10° rotation and 10 Å shift, for a final GSFSC resolution of 4.03 Å before sharpening.

The large *O.i.-raiA* dataset processing is outlined in supplemental Figure S7. In brief, 16,183 movies were imported into cryoSPARC 4.5.1 and motion-corrected and CTF estimated using cryoSPARC Live. After removing 3,749 movies, particles were picked using a general Topaz model, and extracted from 12,405 exposures for a total of 2,362,623 particles. These particles were subjected to three rounds of consecutive heterogeneous refinement (8 classes) using the global *O.i.-raiA* density from the smaller dataset as the initial reference volume. The -1.3 million particles were then refined and re-extracted (216 pixels, 1.70 Å/pixel) and subjected to another three rounds of heterogeneous refinement (5-classes) using the global reconstruction

downsampled to five different resolutions. Particles assigned to classes corresponding to full intact *O.i.-raiA* were then refined, and re-extracted (512 pixels, 0.848 Å/pixel) a final time. These particles were passed through non-uniform refinement and CTF correction for a final global resolution of 2.99 Å.

After refinement, signal corresponding to *O.i.* was subtracted from the particle images, and then the particles for *raiA* were locally refined to a resolution of 3.0 Å. The subtracted particle images were then re-centered on and cropped to a box size of 384 pixels around the *raiA* insert. These particles were then subjected to several rounds of heterogeneous refinement to remove 3D classes with poor density for the *raiA* insert, and the final set of 308,340 particles were put through non-uniform refinement for a final reconstruction with a GSFSC resolution of 3.04 Å.

## Model building and structure refinement

As a starting point for the model building of *O.i.*, 4DS6 structure coordinates were refined in real space by PHENIX[46,47]. The same process was repeated for the locally refined *O.i.*-TPP focused on the scaffold. For the locally refined map focused on the TPP riboswitch, 2GDI structure coordinates were real space refined by PHENIX. Any nucleotide that need to be remodeled was done in COOT[48,49] using the RCrane plugin[50,51]. For the *raiA* motif structure, no starting model was used. The RNA was built de novo in COOT using the secondary structure from Soares, L.W. et al. The final model was then real space refined in PHENIX. UCSF Chimera was used to make figures depicting the cryo-EM density[52]. All software was compiled by SBGrid[53].

## Quantification and statistical analysis

RNA concentrations were determined using a Nanodrop spectrophotometer (Thermo-Fisher). Maturase protein concentrations were determined using an SDS-PAGE gel with a titration of BSA (Thermo-Fisher). To calculate per-reside Full RMSD values, the two models being compared were superposed in COOT using LSQ using the entire sequence for alignment. The superposed models were opened in UCSF Chimera for evaluation using Match -> Align. All map and model validation and statistics were done in PHENIX (Table S1).

## Reporting summary

Further information on research design is available in the Nature Portfolio Reporting Summary linked to this article.

## Data availability

The data supporting the findings of this study are available from the corresponding authors upon request. Structure coordinates have been deposited in the Protein Data Bank (PDB) under accession numbers 9C6I (*O.i.*), 9C6J (*O.i.*-TPP focused on *O.i.*), 9C6K (*O.i.*-TPP focused on TPP) and 9CXF (*O.i.-raiA*). Cryo-EM density maps have been deposited in the Electron Microscopy Data Bank (EMDB) under accession numbers 45247 (*O.i.*), 45248 (*O.i.*-TPP focused on *O.i.*), 45249 (*O.i.*-TPP focused on TPP), 45250 (*O.i.*-TPP apo focused on TPP), 45251 (*T.el.-O.i.* focused on *O.i.*), 45994 (*O.i.-raiA* small data set), and 45988 (*O.i.-raiA* large data set).

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

## Acknowledgements

We would like to thank the Cal-Cryo Facility at UC Berkeley and Daniel Toso for assistance with data collection. Some of this work was performed at the Stanford-SLAC Cryo-EM Center (S2C2), which is supported by the National Institutes of Health Common Fund Transformative High-Resolution Cryo-Electron Microscopy program (U24 GM129541). This work was funded by National Institutes of Health grant R35GM141706 (N.T.) and National Science Foundation grant MCB-2027701 (K.M.W.). The authors would also like to thank the following S2C2 personnel for their invaluable support and assistance: Patrick Mitchell and Megan Louise Mayer. We also thank Jason R. Stagno and Lixin Fan from the National Institutes of Health for providing the SAXS data.

## Author contributions

Conceptualization: D.B.H., B.R., and N.T. Methodology: D.B.H., B.R., S.J., A.K., K.M.W., and N.T. Investigation: D.B.H., B.R., S.J., A.K., K.M.W., and N.T. Visualization: D.B.H., B.R., S.J., A.K., K.M.W., and N.T. Supervision: K.M.W. and N.T. Writing—original draft: D.B.H., B.R., and N.T. Writing—review & editing: D.B.H., B.R., K.M.W., and N.T.

## Competing interests

A provisional patent application has been filed on the use of the group II intron as the scaffolding technology described here with D.B.H., B.R., and N.T. listed as co-inventors. D.B.H., K.M.W., and N.T. are co-founders of A-Form Solutions; N.T. is a co-founder of Syrna Therapeutics; K.M.W. is an advisor to and holds equity in ForagR Medicines and Ribometrix. All other authors declare they have no competing interests.
