## [Transparent Peer Review file · Nature Communications]

Scaffold-enabled high-resolution cryo-EM structure determination of RNA

Corresponding Author: Professor Navtej Toor

Version 0:

Reviewer comments:

Reviewer #1

(Remarks to the Author)
NCOMMS-24-41049

Summary.

The manuscript by Haack et al. proposes a new strategy for high-resolution protein-free RNA structural studies. Leveraging favorable molecular properties of the *Oceanobacillus iheyensis* group IIC intron, the authors append a small RNA riboswitch (TPP) to a rigid linker near domain III to reveal an open and compact TPP conformation at moderate and high resolution, respectively. Complementary SHAPE-MaP analysis supports their conclusions defining an exciting avenue for future RNA structural studies. The quality of the data is high, and the manuscript is well-written. This work has potential significance for the RNA and structural biology communities, as well as, the field of RNA therapeutics.

Please find specific comments below.

The authors make a compelling case that the group IIC intron is a powerful strategy for overcoming the significant barriers associated with cryoEM studies of the small TPP RNA. However, the versatility of this approach is unclear. Did the authors apply this approach to another small RNA to show how well the group IIC intron can be adapted to a distinct structured RNA? This follow-up is necessary to show there is potential in its broader applicability.

The authors unveil a new Y-shaped TPP structure that was not reported previously. However, at 6 Å resolution, the molecular details of this novel state remain a mystery. Did the authors try collecting more data in the absence of a ligand to improve the resolution of the ligand-free structure? Half as many movies were recorded in the ligand-free dataset compared to the high-resolution dataset with ligand.

The authors should include a paragraph in the discussion describing general recommendations for prospective users. For example, what are important RNA design considerations, what are potential limitations when studying more complex systems, what are the important image processing considerations, and what validation methods will be required when studying chimeric molecules?

Can the authors comment on why the group IIC intron may outperform the group IIB intron? Are there general lessons that can inform future scaffold development?

Reviewer #2

(Remarks to the Author)
Key Results

The authors describe the development of a group II intron scaffold that facilitates cryo-EM structure determination of attached small RNAs at high resolution. They successfully applied this scaffold to solve cryo-EM structures of the thiamine pyrophosphate (TPP) riboswitch in both ligand-bound and ligand-free states. While the use of scaffolds for RNA structure determination has been explored before, the determination of the apo structure of the TPP riboswitch represents a novel achievement.

Validity and Significance

1. The statement on page 9 that the authors' "scaffold strategy is the first to allow cryo-EM structure determination of a target RNA to better than 3 Å resolution" is misleading. It could be interpreted as a general claim that the group II intron scaffold allows for the determination of RNA structures universally at this resolution, which is not supported by the data. Moreover, it incorrectly implies that other scaffolding approaches previously reported do not allow achieving similar resolutions. Although this work does report the first scaffold-based RNA structure at a resolution better than 3 Å, generalizing from a single structure to the overall capability of the methodology is an overstatement.

The manuscript lacks a comprehensive comparison of the authors' approach with existing methodologies. Although the authors cite the work of Langeberg et al., Liu et al., and Kappel et al., they omit reference to recent relevant work, such as Sampedro Vallina et al. (2023), who developed RNA origami scaffolds for cryo-EM. A detailed comparison, highlighting both the strengths and limitations of the group II intron scaffold relative to other methods/scaffolds, is necessary. This should include considerations such as the size, stability, and ease of production of RNA scaffolds with the inserted RNAs of interest. The authors could include a table or figure that systematically compares their scaffold with other scaffolding techniques in terms of key parameters such as resolution achieved in published studies and the potential resolution limit (as just one or a few studies are not sufficient as a general indicator of a resolution of a method), applicability to different RNA types, ease of use, etc. This would provide a clearer, more objective assessment of where their work fits within the broader field.

Furthermore, the manuscript does not sufficiently address the limitations of the group II intron scaffold, such as its applicability to different types of RNAs or potential size constraints.

2. The authors propose a mechanistic model for the TPP riboswitch based on the structures they have determined. However, this model includes the structure and interactions with the expression platform of the riboswitch, which was not experimentally analyzed in this study. Given that similar models have been proposed previously, the new mechanistic insights provided by this work are limited. The manuscript also lacks a comparison with previously published TPP riboswitch structures and does not adequately credit prior mechanistic studies on this riboswitch.

If the group II intron scaffold is as powerful as the authors suggest, it would be valuable to also determine the structures of the ligand-bound and apo forms of the TPP riboswitch, including the expression platform. This would allow for a more complete validation of the proposed molecular model and would provide support for the applicability of the scaffold.

Data and Methodology

The methodology presented is generally sound, but the level of detail is insufficient for full reproducibility. In addition, the manuscript lacks detailed validation of the structures, particularly regarding stereochemical parameters and the assessment of model fit to the cryo-EM map, such as using Q-scores.

Minor Remark

Page 1: The statement "We have found that vitrification on cryo-EM grids often results in the denaturation and aggregation of RNA" suggests that this phenomenon is being reported for the first time. This should be corrected to acknowledge prior studies that have observed and reported similar challenges in cryo-EM of RNA.

Summarizing, additional experiments and analyses, particularly a more in-depth study of the TPP riboswitch with additional experimental data, are needed to support the authors' claims.

In addition to the specific comments listed above: The authors should better articulate the novelty of their approach in the context of existing literature. They should clearly define what sets their method apart from prior scaffolding techniques and why their specific scaffold might be more broadly applicable or beneficial for cryo-EM studies of small RNAs. The authors should include more specifics on the preparation and validation steps, such as detailed protocols for RNA preparation, cryo-EM grid preparation, and data processing. This will not only help in reproducibility but also in the transparency of the study. The authors should include a more in-depth discussion of the limitations of their scaffold, both in terms of its general applicability to other RNAs and any potential challenges they foresee with this approach. This could involve addressing the range of RNAs it can handle and any potential issues when using this scaffold. Last but not least, I recommend that the authors review their language, particularly in statements about being the "first" or "only" to achieve certain milestones or to allow something. They should avoid overstatement and ensure that their claims are well-supported by the data and literature.

Reviewer #3

(Remarks to the Author)

In this manuscript, Haack and collaborators present a novel method for enhancing the structure determination of smaller RNA particles at higher resolutions using cryo-EM. The study begins with the observation that most RNA structures

deposited in the PDB are resolved at moderate resolutions, especially compared to protein structures. To address this limitation, the authors propose a strategy to enhance the stability of RNA macromolecules by fusing them with group II introns. These introns are large, compact ribozymes that can serve as scaffolds, stabilizing the RNA and making it more suitable for high-resolution structural determination. The group II C intron was identified as a suitable scaffold for the structural determination of a fused RNA molecule. The authors demonstrate this approach by stabilizing the thiamine pyrophosphate (TPP) riboswitch, which is less than 100 nucleotides in length, through its attachment to the domain III stem of group IIC introns. This fusion allowed the determination of the riboswitch structure at a resolution of 2.5 Å. Furthermore, they successfully resolved the TPP riboswitch in both its apo and ligand-bound states, providing extended functional insights into this molecule. The work is of significance for the cryo-EM and RNA fields. The main claim of the manuscript, namely the improvement of RNA structures resolution by fusion with the group II intron scaffold, is well supported experimentally.

The manuscript can be considered for publication in Nature Communications upon addressing the following points:

1. The authors present the ligand-free structure of the TPP riboswitch, where the sensing stems are nearly perpendicular to each other, contrasting with the compact conformation observed in the TPP-bound state. They interpret this structure as representing a physiological state that precedes TPP binding, wherein a regulatory stem-loop interacts with the riboswitch (as depicted in Figure 6B). However, the authors do not cite any references supporting the existence of this regulatory element adjacent to the riboswitch. It is unclear whether this evidence has roots in the published literature, or if this element was detected exclusively in this paper (by sequence analysis or other means?). This ambiguity must be thoroughly addressed and clarified in greater detail.

Additionally, without further supporting evidence, the proposed model appears somewhat speculative. While it is acceptable to present this model, the manuscript should explicitly state that it is hypothetical and based on limited data.

2. The cryo-EM structure of the TPP-apo riboswitch fused to group II intron differs from the previously reported crystal structure, potentially due to crystal packing effects. To ensure that the fusion with group II introns does not alter the fold of the TPP-apo riboswitch, it is crucial to rule out the possibility that the intron fusion itself is preventing the closure of the apo form. The complete density corresponding to the group II introns fused to the TPP-apo riboswitch should be clearly shown. Besides, a tentative structural model should be constructed in the density, even at a lower resolution, such as 6 Å. Existing structural data of individual modules can facilitate this modelling.

Additionally, the authors mention that the SAXS reconstruction of the ligand-free riboswitch does not align with the previously published crystal structure. It is important to clarify whether the SAXS reconstruction matches the cryo-EM structure resolved in this study. A comparison between the SAXS model and the cryo-EM density should be included to support this observation.

Minor points:

The upper four structures from Figure 1 would be better suited as supplementary figures, as they do not pertain to data generated in the current manuscript. Additionally, the zoomed-in insets should be enlarged to enhance clarity.

In Figure 2, the density corresponding to the embedded RNA should be explicitly labeled as belonging to the IIC intron.

Furthermore, Figure 2B requires additional labels for clarity.

In Figure 4C, the low-threshold map is difficult to discern. Improving the contrast would make the map clearer.

For Figure 5A, the structure of the TPP riboswitch and its assigned structural elements should be clearly indicated.

In Figure 6, the various structural elements (P1, P3, P4, L3, L5) do not correspond with the diagram presented in Figure 4A, which features elements P2-P5 and L5. This inconsistency should be addressed.

In the Discussion, the statement referring to "The thiamine sensing stem in the ligand-free state...an additional helical groove (Fig. 5A)" seems inaccurate, as there is no depiction of the ligand-free state in Figure 5A.

Version 1:

Reviewer comments:

Reviewer #1

(Remarks to the Author)
NCOMMS-24-41049A

Summary:

The revised manuscript by Haack et al. incorporates substantial improvements. Specifically, the authors demonstrate the potential for broad applicability of small RNA analysis by analyzing the scaffold *raiA* ncRNA. The authors also include detailed guidance on important considerations and limitations when using the group IIC intron for cryoEM studies. These amendments have adequately addressed my previous concerns.

Reviewer #2

(Remarks to the Author)

The authors have addressed all of my comments, questions, and suggestions from the initial review. In this revised version of the manuscript, they have included an additional structure determined using their approach with the group II intron scaffold, for *raiA* ncRNA, which is much bigger than the TPP riboswitch presented in the original submission.

The authors describe this *raiA* structure as the first and unique structure available for *raiA*. However, a structure of *raiA* was recently published (PDB ID 9G7C, <https://www.sciencedirect.com/science/article/pii/S0022283624004625>), at a lower resolution of approximately 5 Å.

I suggest that the authors revise sections of the manuscript that discuss the *raiA* structure to remove claims of novelty, priority, and the absence of an *raiA* structure in the PDB, such as “*RaiA* is a new class of structured RNAs that has no known homologues in the Protein Data Bank” in the abstract, “there are no homologues in the Protein Data Bank (PDB)” in the introduction, “*RaiA* RNA as a target of unknown structure” in the Results section, and “the *raiA* motif, an RNA with no structural homolog” in the Discussion. Instead, the availability of this other *raiA* structure, determined without scaffolding and at much worse resolution, provides an excellent opportunity for the authors to highlight the advantages of their approach.

For example, in the previously published 5 Å structure, only a small fraction of nucleotides are resolved. In contrast, in the new 2.6 Å structure from this work, cryo-EM density is visible for every nucleotide, and in many cases purines and pyrimidines can be distinguished. This strongly demonstrates the utility of the scaffolding approach.

Reviewer #3

(Remarks to the Author)

Upon reviewing the revised manuscript, I find that the authors have comprehensively addressed each of the points raised during the initial review process. The modifications have enhanced the manuscript, particularly in terms of clarity and depth of the discussion. The manuscript is suitable for publication in *Nature Communications*.

RESPONSE TO REFEREES:

My colleagues and I would like to thank the referees for their careful analysis of our work and their thoughtful comments, which have been very helpful in revising the manuscript. Our responses to specific comments are enumerated below:

Reviewers' Comments:

Reviewer #1 (Remarks to the Author):

NCOMMS-24-41049

Summary.

The manuscript by Haack et al. proposes a new strategy for high-resolution protein-free RNA structural studies. Leveraging favorable molecular properties of the *Oceanobacillus iheyensis* group IIC intron, the authors append a small RNA riboswitch (TPP) to a rigid linker near domain III to reveal an open and compact TPP conformation at moderate and high resolution, respectively. Complementary SHAPE-MaP analysis supports their conclusions defining an exciting avenue for future RNA structural studies. The quality of the data is high, and the manuscript is well-written. This work has potential significance for the RNA and structural biology communities, as well as, the field of RNA therapeutics.

Please find specific comments below.

The authors make a compelling case that the group IIC intron is a powerful strategy for overcoming the significant barriers associated with cryoEM studies of the small TPP RNA. However, the versatility of this approach is unclear. Did the authors apply this approach to another small RNA to show how well the group IIC intron can be adapted to a distinct structured RNA? This follow-up is necessary to show there is potential in its broader applicability.

We agree with this comment. We have applied this methodology to a newly discovered bacterial non-coding RNA of unknown structure called *raiA*. The cryo-EM structure of this 210-nucleotide RNA has been determined to 2.9 Å. This shows applicability to another distinct structured RNA.

The authors unveil a new Y-shaped TPP structure that was not reported previously. However, at 6 Å resolution, the molecular details of this novel state remain a mystery. Did the authors try collecting more data in the absence of a ligand to improve the resolution of the ligand-free structure? Half as many movies were recorded in the ligand-free dataset compared to the high-resolution dataset with ligand.

The riboswitch construct exhibited altered solution behavior in the absence of the ligand leading to less particles in grid holes. As a result, we obtained significantly fewer particles per micrograph for the apo Y state. To overcome this issue, we anticipate that

collecting a cryo-EM dataset about an order of magnitude larger than for the bound state would result in high resolution. In addition, the more dynamic nature of the apo state also increases the requirement for a larger dataset. This is beyond our current research budget for the scope of this study.

The authors should include a paragraph in the discussion describing general recommendations for prospective users. For example, what are important RNA design considerations, what are potential limitations when studying more complex systems, what are the important image processing considerations, and what validation methods will be required when studying chimeric molecules?

We added two sections to the Results section entitled “Construct design to attach target to the scaffold” and “Cryo-EM Data Processing of Scaffolded RNAs” to address this request.

Can the authors comment on why the group IIC intron may outperform the group IIB intron? Are there general lessons that can inform future scaffold development?

We are not certain why the group IIC intron outperforms both the group IIB and group I introns as a high-resolution scaffold. The orientation distribution is more uniform for the group IIC intron and we hypothesize that might make this particular scaffold more resilient to any negative effects of the attachment of a target RNA.

Reviewer #2 (Remarks to the Author):

Key Results

The authors describe the development of a group II intron scaffold that facilitates cryo-EM structure determination of attached small RNAs at high resolution. They successfully applied this scaffold to solve cryo-EM structures of the thiamine pyrophosphate (TPP) riboswitch in both ligand-bound and ligand-free states. While the use of scaffolds for RNA structure determination has been explored before, the determination of the apo structure of the TPP riboswitch represents a novel achievement.

Validity and Significance

1. The statement on page 9 that the authors' "scaffold strategy is the first to allow cryo-EM structure determination of a target RNA to better than 3 Å resolution" is misleading. It could be interpreted as a general claim that the group II intron scaffold allows for the determination of RNA structures universally at this resolution, which is not supported by the data. Moreover, it incorrectly implies that other scaffolding approaches previously reported do not allow achieving similar resolutions. Although this work does report the

first scaffold-based RNA structure at a resolution better than 3 Å, generalizing from a single structure to the overall capability of the methodology is an overstatement.

The manuscript lacks a comprehensive comparison of the authors' approach with existing methodologies. Although the authors cite the work of Langeberg et al., Liu et al., and Kappel et al., they omit reference to recent relevant work, such as Sampedro Vallina et al. (2023), who developed RNA origami scaffolds for cryo-EM. A detailed comparison, highlighting both the strengths and limitations of the group II intron scaffold relative to other methods/scaffolds, is necessary. This should include considerations such as the size, stability, and ease of production of RNA scaffolds with the inserted RNAs of interest. The authors could include a table or figure that systematically compares their scaffold with other scaffolding techniques in terms of key parameters such as resolution achieved in published studies and the potential resolution limit (as just one or a few studies are not sufficient as a general indicator of a resolution of a method), applicability to different RNA types, ease of use, etc. This would provide a clearer, more objective assessment of where their work fits within the broader field.

Furthermore, the manuscript does not sufficiently address the limitations of the group II intron scaffold, such as its applicability to different types of RNAs or potential size constraints.

To avoid any confusion, we have added the following statement: “In summary, current scaffolding approaches have not yet demonstrated high resolution structure determination of attached target RNAs to allow discrimination of individual nucleobases.”

We have deleted the statement including the following phrase: “...scaffold strategy is the first to allow cryo-EM structure determination of a target RNA to better than 3 Å resolution.”

We have now included references to the RNA origami and ribosome scaffold approaches. In our opinion, the only relevant standard for comparing the different scaffolds is the quality of the final cryo-EM map and its resolution. We have referenced the previous scaffold approaches and the fact that none to date have achieved high resolution. A detailed comparison of all RNA scaffold approaches would be more appropriate for a subsequent review.

We have added the high-resolution structure of the *raiA* bacterial non-coding RNA as another successful demonstration of the group II intron scaffold approach. The manuscript now covers the size range from 86 to 210 nucleotides at high resolution.

2. The authors propose a mechanistic model for the TPP riboswitch based on the structures they have determined. However, this model includes the structure and interactions with the expression platform of the riboswitch, which was not experimentally analyzed in this study. Given that similar models have been proposed previously, the new mechanistic insights provided by this work are limited. The manuscript also lacks a

comparison with previously published TPP riboswitch structures and does not adequately credit prior mechanistic studies on this riboswitch.

We already show a comparison between the cryo-EM and x-ray structures of bound TPP (Supplementary Figure S5). The structures are almost identical except for solvent exposed loops which are constrained by the crystal lattice as expected. RMSD values for this comparison are shown in Supplementary Figure S5. We also already compare the apo state with the crystal dimer seen in a crystal structure. We have added more references to mechanistic studies of the TPP riboswitch.

If the group II intron scaffold is as powerful as the authors suggest, it would be valuable to also determine the structures of the ligand-bound and apo forms of the TPP riboswitch, including the expression platform. This would allow for a more complete validation of the proposed molecular model and would provide support for the applicability of the scaffold.

We agree that it would be valuable to obtain a high resolution structure of the riboswitch aptamer with its expression platform. We previously explored this possibility, however we could not identify a stem to connect to the group II intron other than the base P1 stem, as other regions are likely functionally important. Once the expression platform is added, this P1 stem cannot be used for attachment. Our scaffold approach requires the attachment point on the target RNA to be phylogenetically variable so that the function can be retained for a biologically relevant structure.

Data and Methodology

The methodology presented is generally sound, but the level of detail is insufficient for full reproducibility. In addition, the manuscript lacks detailed validation of the structures, particularly regarding stereochemical parameters and the assessment of model fit to the cryo-EM map, such as using Q-scores.

We have provided all validation reports for the deposited structures containing statistics such as Molprobit scores and Q-scores.

Minor Remark

Page 1: The statement "We have found that vitrification on cryo-EM grids often results in the denaturation and aggregation of RNA" suggests that this phenomenon is being reported for the first time. This should be corrected to acknowledge prior studies that have observed and reported similar challenges in cryo-EM of RNA.

We have not found any references stating that RNA is prone to aggregation and denaturation in thin ice. This is based purely on our own observations and is consistent with the lack of deposited high-resolution protein-free RNA structures in the PDB.

Summarizing, additional experiments and analyses, particularly a more in-depth study of the TPP riboswitch with additional experimental data, are needed to support the authors' claims.

We have added references to prior work done on biochemical characterization of the TPP riboswitch. These references were not present in the first draft. This biochemical work is completely consistent with our cryo-EM structures.

In addition to the specific comments listed above: The authors should better articulate the novelty of their approach in the context of existing literature. They should clearly define what sets their method apart from prior scaffolding techniques and why their specific scaffold might be more broadly applicable or beneficial for cryo-EM studies of small RNAs. The authors should include more specifics on the preparation and validation steps, such as detailed protocols for RNA preparation, cryo-EM grid preparation, and data processing. This will not only help in reproducibility but also in the transparency of the study. The authors should include a more in-depth discussion of the limitations of their scaffold, both in terms of its general applicability to other RNAs and any potential challenges they foresee with this approach. This could involve addressing the range of RNAs it can handle and any potential issues when using this scaffold. Last but not least, I recommend that the authors review their language, particularly in statements about being the "first" or "only" to achieve certain milestones or to allow something. They should avoid overstatement and ensure that their claims are well-supported by the data and literature.

We have added a section to the Results in which we discuss construct design considerations when attaching a target RNA to the scaffold. We have detailed protocols for sample preparation and cryo-EM data processing in the Methods section. We include the phrase "first to demonstrate" instead of only "first" to avoid any confusion.

Reviewer #3 (Remarks to the Author):

In this manuscript, Haack and collaborators present a novel method for enhancing the structure determination of smaller RNA particles at higher resolutions using cryo-EM. The study begins with the observation that most RNA structures deposited in the PDB are resolved at moderate resolutions, especially compared to protein structures. To address this limitation, the authors propose a strategy to enhance the stability of RNA macromolecules by fusing them with group II introns. These introns are large, compact ribozymes that can serve as scaffolds, stabilizing the RNA and making it more suitable for high-resolution structural determination. The group II C intron was identified as a suitable scaffold for the structural determination of a fused RNA molecule. The authors demonstrate this approach by stabilizing the thiamine pyrophosphate (TPP) riboswitch, which is less than 100 nucleotides in length, through its attachment to the domain III stem of group IIC introns. This fusion allowed the determination of the riboswitch structure at a resolution of 2.5 Å. Furthermore, they successfully resolved the

TPP riboswitch in both its apo and ligand-bound states, providing extended functional insights into this molecule.

The work is of significance for the cryo-EM and RNA fields. The main claim of the manuscript, namely the improvement of RNA structures resolution by fusion with the group II intron scaffold, is well supported experimentally.

The manuscript can be considered for publication in Nature Communications upon addressing the following points:

1. The authors present the ligand-free structure of the TPP riboswitch, where the sensing stems are nearly perpendicular to each other, contrasting with the compact conformation observed in the TPP-bound state. They interpret this structure as representing a physiological state that precedes TPP binding, wherein a regulatory stem-loop interacts with the riboswitch (as depicted in Figure 6B). However, the authors do not cite any references supporting the existence of this regulatory element adjacent to the riboswitch. It is unclear whether this evidence has roots in the published literature, or if this element was detected exclusively in this paper (by sequence analysis or other means?). This ambiguity must be thoroughly addressed and clarified in greater detail. Additionally, without further supporting evidence, the proposed model appears somewhat speculative. While it is acceptable to present this model, the manuscript should explicitly state that it is hypothetical and based on limited data.

We have now added key references to the prior work that first postulated the model for TPP function with the regulatory element. We have also added references for this hypothesis and supporting biochemical data.

2. The cryo-EM structure of the TPP-apo riboswitch fused to group II intron differs from the previously reported crystal structure, potentially due to crystal packing effects. To ensure that the fusion with group II introns does not alter the fold of the TPP-apo riboswitch, it is crucial to rule out the possibility that the intron fusion itself is preventing the closure of the apo form. The complete density corresponding to the group II introns fused to the TPP-apo riboswitch should be clearly shown. Besides, a tentative structural model should be constructed in the density, even at a lower resolution, such as 6 Å. Existing structural data of individual modules can facilitate this modelling.

Fig. RF1 shows the complete density of the group II intron fused to the TPP-apo riboswitch. The density show that there is no contact between the apo Y-shape and the group II intron. Our interpretation of the density of the thiamine sensing stem is that it undergoes a rearrangement that cannot be accurately modeled at this low resolution. Any modeling done at this resolution would be speculative and might be misleading to researchers that would download the resulting PDB.

Additionally, the authors mention that the SAXS reconstruction of the ligand-free riboswitch does not align with the previously published crystal structure. It is important to clarify whether the SAXS reconstruction matches the cryo-EM structure resolved in this study. A comparison between the SAXS model and the cryo-EM density should be included to support this observation.

The authors of that previously published crystal structure provided us with the SAXS data. We have included a comparison of the SAXS data with our apo structure in Fig. RF2 (next page). Our apo Y-shaped structure exhibits a good fit within the SAXS envelope. The helix extending out of the enveloped represents the extra sequence used to attach the riboswitch to the scaffold.

Minor points:

The upper four structures from Figure 1 would be better suited as supplementary figures, as they do not pertain to data generated in the current manuscript. Additionally, the zoomed-in insets should be enlarged to enhance clarity.

Figure 1 is now Figure S1

In Figure 2, the density corresponding to the embedded RNA should be explicitly labeled as belonging to the IIC intron. Furthermore, Figure 2B requires additional labels for clarity.

This change has been made.

In Figure 4C, the low-threshold map is difficult to discern. Improving the contrast would make the map clearer.

Contrast has been improved.

For Figure 5A, the structure of the TPP riboswitch and its assigned structural elements should be clearly indicated.

This change has been incorporated into our updated figures.

In Figure 6, the various structural elements (P1, P3, P4, L3, L5) do not correspond with the diagram presented in Figure 4A, which features elements P2-P5 and L5. This inconsistency should be addressed.

This change has been incorporated into our updated figures.

In the Discussion, the statement referring to “The thiamine sensing stem in the ligand-free state...an additional helical groove (Fig. 5A)” seems inaccurate, as there is no depiction of the ligand-free state in Figure 5A.

This state is labeled as “apo” in the figure, which is a synonym for “ligand-free” state. We have changed the text to refer to “apo” to avoid confusion.